# Lysosome-related organelles promote stress and immune responses in *C. elegans*

Gábor Hajdú [1], Milán Somogyvári[1], Péter Csermely [1] & Csaba Sőti [1]✉

Lysosome-related organelles (LROs) play diverse roles and their dysfunction causes immunodeficiency. However, their primordial functions remain unclear. Here, we report that *C. elegans* LROs (gut granules) promote organismal defenses against various stresses. We find that toxic benzaldehyde exposure induces LRO autofluorescence, stimulates the expression of LRO-specific genes and enhances LRO transport capacity as well as increases tolerance to benzaldehyde, heat and oxidative stresses, while these responses are impaired in *glo-1*/Rab32 and *pgp*-2 ABC transporter LRO biogenesis mutants. Benzaldehyde upregulates *glo-1*- and *pgp-2*-dependent expression of heat shock, detoxification and antimicrobial effector genes, which requires *daf-16*/FOXO and/or *pmk-1*/p38MAPK. Finally, benzaldehyde preconditioning increases resistance against *Pseudomonas aeruginosa* PA14 in a *glo-1*- and *pgp-2*-dependent manner, and PA14 infection leads to the deposition of fluorescent metabolites in LROs and induction of LRO genes. Our study suggests that LROs may play a role in systemic responses to stresses and in pathogen resistance.

[1] Department of Molecular Biology, Semmelweis University, Budapest, Hungary. ✉email: soti.csaba@med.semmelweis-univ.hu

Lysosome-related organelles (LROs) are members of the endolysosomal family of cell type-specific intracellular compartments[1,2]. LROs share several features with canonical lysosomes, such as endosomal lineage, acidic milieu, and hydrolases, as well as lysosome-specific membrane proteins[1,2]. In humans, LROs participate in major physiological processes *via* storage and controlled secretion of material critical for tissue-specific functions. Such examples include the blood clotting factors of platelet delta-granules, melanin of skin pigment cell melanosomes, various granules of polymorphonuclear and mononuclear blood cells involved in immune response, and surfactant-producing lamellar bodies of pulmonary alveolar cells[2,3]. Mutations in genes crucial for LRO biogenesis are implicated in several inherited rare disorders developing co-existing symptoms of hypopigmentation, immunodeficiency, bleeding diathesis, and neuropathy (e.g., Hermansky-Pudlak Syndrome, Chediak-Higashi Syndrome, Griscelli Syndrome)[4,5]. The diverse function and severe diseases in the absence of LROs suggest a general role in self-protection at the organism–environment interface.

Primordial LROs appear early in evolution as acidocalcisomes of unicellular organisms, *C. elegans* gut granules, *D. melanogaster* pigment granules, and *D. rerio* silver pigment granules[6–9]. In genetic *C. elegans* screens numerous *glo* (gut granule *lo*ss) genes emerged as they were critical for LRO formation and systematically characterized by the Hermann lab[7,10–14]. The mammalian *glo* gene ortholog RAB32/38 GTPases direct the biogenesis of melanosomes and other LROs, for instance, together with its guanine nucleotide exchange factor BLOC-3, they facilitate the recycling of biogenetic and fusion components from melanosomes[15,16]. Additionally, members of the ABC-transporter family, endosomal vesicular transport proteins as well as components of HOPS complex and endolysosomal fusion proteins were demonstrated to be involved in LRO biogenesis and function independently of endosome-lysosome fusion and apoptotic clearance[12–14,17]. *C. elegans* gut granules contain birefringent and autofluorescent materials that accumulate during aging, making them a marker of healthspan[7,18–20]. Different sources of autofluorescence were identified, such as advanced glycation end products of proteins fluorescing in red, as well as anthranilic acid glucoside conjugates, which at the time of death were released from LROs and exhibited an anteroposterior wave of blue fluorescence burst to visualize organismal demise[20,21]. Furthermore, similar to *D. melanogaster*[22], *C. elegans* LROs were demonstrated as active zinc storage sites which required the *cdf-2* cation transporter[23]. It was also reported that LRO mutants exhibit reduced lifespan and impaired resistance to proteotoxicity[24]. A recent systematic study showed that the synthesis of a plethora of ascarosides and glucosides also relies on *glo-1* and intestinal carboxylesterases, probably in LROs[25]. Nonetheless, the biological function of *C. elegans* LROs is still enigmatic.

Employing a toxic stress paradigm developed in our lab[26] we observed a robust increase of autofluorescence in gut granules in toxic odorant-exposed worms. We found that upon toxic insults, LROs are required to stimulate the expression of cytoprotective stress, detoxification, and immune effector genes and confer organismal protection against various stresses and pathogen infection. The LRO-associated expression of stress genes mainly requires the *daf-16* FOXO and/or the *pmk-1* p38 MAPK orthologs and yet unknown factors. These findings establish a role for *C. elegans* LROs in systemic stress and immune defenses.

## Results

### Toxic benzaldehyde exposure induces a rapid accumulation of autofluorescent material in *C. elegans* LROs.
The volatile benzaldehyde (BA) forms protein adducts, disrupts membranes, and causes oxidative stress[27]. Although at low concentrations it is an attractive neuronal nutrient signal for *C. elegans*, at high doses it is toxic where it activates a widespread, but specific stress and detoxification response in non-neuronal tissues[26]. Upon exposure to undiluted (i.e., 100% v/v) BA for 4 h a strong granular autofluorescence was observed in 1-day young adult worms (Fig. 1a). This was reminiscent of the autofluorescent material which appears in intestinal LROs of aged worms. The vital dye Nile Red was demonstrated to be taken up by LROs in *C. elegans*[10]. As BA-induced autofluorescence was not observed in the red channel (Supplementary Fig. 1a), we conducted Nile Red feeding experiments which showed a co-localization of the green autofluorescence with red Nile Red fluorescence in the intestine (Fig. 1b). The *glo-1* gene encodes the ortholog of human Rab32 GTPase and its loss of function mutants are defective in the formation of gut granules[7,12]. Hence, *glo-1* mutants were treated with BA and exhibited a complete absence of both basal and BA exposure-induced intestinal autofluorescence (Fig. 1c) and they neither exhibited Nile Red fluorescence (Supplementary Fig. 1b). Autofluorescent materials accumulating during aging include the glucosyl ester conjugate of anthranilic acid (AA), a metabolite of tryptophan, which has a predominantly blue fluorescence[20,28,29]. Similarly to that reported to AA, the BA-induced autofluorescence was very intense in the blue (DAPI) channel, less intense, and easily detectable in the green (GFP) channel, whereas entirely absent in red (Supplementary Figs. 1a and 2a). However, BA-treated young *kynu-1* mutants deficient in the production of AA[28,30] exhibited an increase in autofluorescence similar to wild type (Supplementary Fig. 1c). These findings suggest that the material observed in the intestine in response to BA does not stimulate AA formation and the fluorescent material is independent of AA-derived products.

Although we found that undiluted (i.e., 100% v/v) diacetyl (DA) is also highly toxic[26], neither DA nor the strong repellent nonanone elevated the autofluorescence signal (Supplementary Fig. 1d). While DA toxicity did not induce apparent cytoprotective responses, treatment with undiluted (i.e., 100% v/v) methyl-salicylate (MS), an odorant chemically closely related to BA stimulated identical molecular defenses, including DAF-16 translocation as well as activation of the *gst-4* and *cyp-35B* promoters[26]. Hence, we asked whether exposure to MS might induce autofluorescence, and it robustly did (Supplementary Fig. 1d). Thus, we conclude that exposure to BA and MS elicits the accumulation of unknown autofluorescent material(s) and/or the chemical modification of pre-existing material in intestinal LROs.

### LRO biogenetic genes are required for preconditioning-induced tolerance to BA toxicity.
We have previously shown that BA preconditioning confers protection against a lethal BA exposure by engaging a *daf-16-* and *skn-1*-dependent stress- and detoxification response[26]. Based on this finding we hypothesized that the sequestration, breakdown, or conversion of BA to autofluorescent material by LROs might contribute to the organismal defense against BA toxicity. To investigate this, we estimated the survival of wild-type and *glo-1* mutant nematodes exposed to a lethal dose of BA, after hormetic BA preconditioning (PC) (Fig. 2a). Untreated *glo-1* animals exhibited similar survival to wild type, suggesting that BA per se does not induce severe pathology in the absence of *glo-1* (Fig. 2b). However, BA PC significantly improved the physiological tolerance to BA in wild-type, but not in *glo-*1 mutants (Fig. 2b). The similar basal and impaired preconditioning-induced stress tolerance is in agreement with the phenotype of other stress response mutants[31].

The *glo-3* gene, together with *ccz-1* is likely to function as the guanine nucleotide exchange factor (GEF) of the GLO-1/Rab32 GTPase, facilitating GLO-1 association with gut granules[12]. Adult *glo-3* mutants contain a reduced number of intestinal LROs,

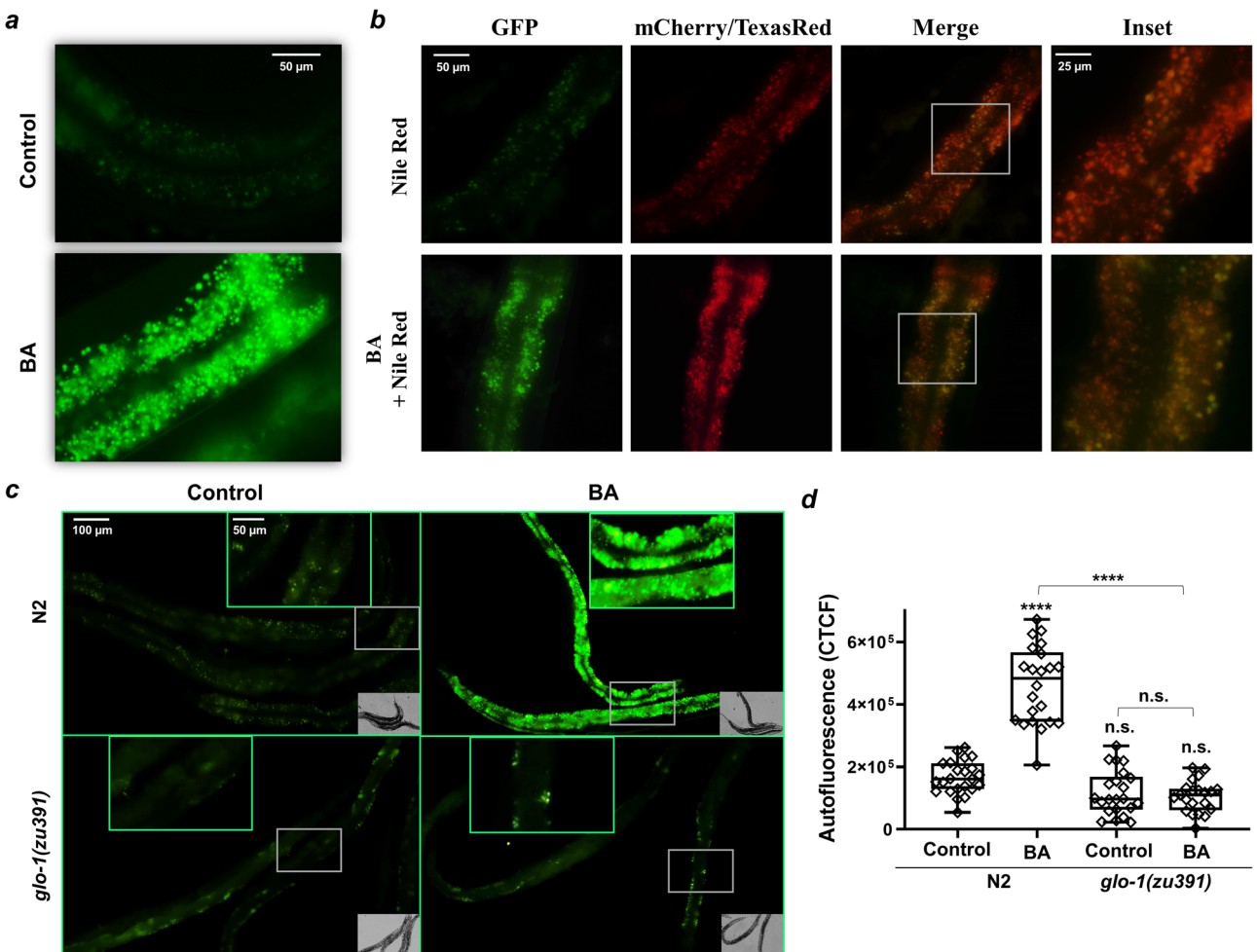

**Fig. 1 Toxic benzaldehyde (BA) exposure induces a rapid accumulation of autofluorescent material in LROs. a** Fluorescence microscopy images of BA-exposed wild-type worms. **b** Fluorescence microscopy images of BA-exposed wild-type worms simultaneously fed with Nile Red. **c** Fluorescence microscopy images of BA-exposed wild-type and *glo-1(zu391)* mutants. Animals were exposed to 1 µl of undiluted BA for 4-h. Images represent three independent experiments with similar results. **d** Quantification of the data from panel **c** from three independent experiments using at least 20 animals per condition. Boxes represent median and first and third quartiles and whiskers represent 10th to 90th percentiles. *p*-values were obtained by one-way ANOVA with Fisher's LSD post hoc test following the evaluation of normal distribution significance by the Kolmogorov–Smirnov test (f). n.s., not significant; ****p < 0.0001.

which are stained by Nile Red, but do not contain GLO-1 protein and do not accumulate birefringent material[13]. To decipher whether these LROs may function to protect from BA toxicity, we took use of two *glo-3* alleles[13]. Both the Class II *glo-3(zu446)* and the Class I *glo-3(kx94)* alleles largely eliminated the BA-induced intestinal autofluorescence (Fig. 2c). Likewise, both of them also prevented the BA PC-induced increase in survival against a lethal BA stress, although the *glo-3(kx94)* mutant showed a partial effect, for which the reason is unknown (Fig. 2d). Nevertheless, these results suggest that not only the presence of gut granules, but their proper maturation and quantity are required for the accumulation and/or modification of autofluorescent material upon BA exposure and for sufficient protection against BA-induced toxic stress.

*flu-1* mutants are defective in the catabolism of AA and accumulate more blue fluorescent material in LROs compared to wild type[7,32]. We, therefore, asked how the load of LROs with AA in *flu-1* worms interferes with the events elicited by BA exposure. We observed an increased basal autofluorescence in *flu-1* compared to the wild type in the green and especially in the blue channel, characteristic of AA (Supplementary Fig. 2a)[28,32]. However, upon BA treatment, fluorescence of *flu-1* worms was

not increased as much as that of the wild type, which showed an increased signal in both green and blue channels (Supplementary Fig. 2a). Furthermore, *flu-1* animals were deficient in both basal and BA PC-induced physiological tolerance against BA toxicity (Supplementary Fig. 2b). Hence, either the chronic AA overload in *flu-1* worms might impair LRO function or perhaps *flu-1* might be partially involved in BA-induced autofluorescence and in the adaptive response to BA toxicity. Based on our finding of increased gut granule autofluorescence after MS exposure, we also preconditioned nematodes with MS, and observed an increased survival after lethal BA exposure (Supplementary Fig. 2c). In contrast, neither BA nor MS PC influenced the survival in response to the structurally unrelated DA, which did not affect LRO autofluorescence (Supplementary Fig. 2c). These observations further suggest a role for chemical structure-specific LRO-associated processes in handling BA toxicity.

Previously, we showed that the efficiency of cytoprotective responses regulates behavioral responses to toxic stress[26]. Therefore, we asked whether the induction of physiological stress tolerance influences behavioral adaptation to BA exposure *via* LRO-mediated processes. We examined the lawn leaving behavior of wild-type and *glo-1* mutants in the presence of BA, after BA PC (Fig. 2e). As

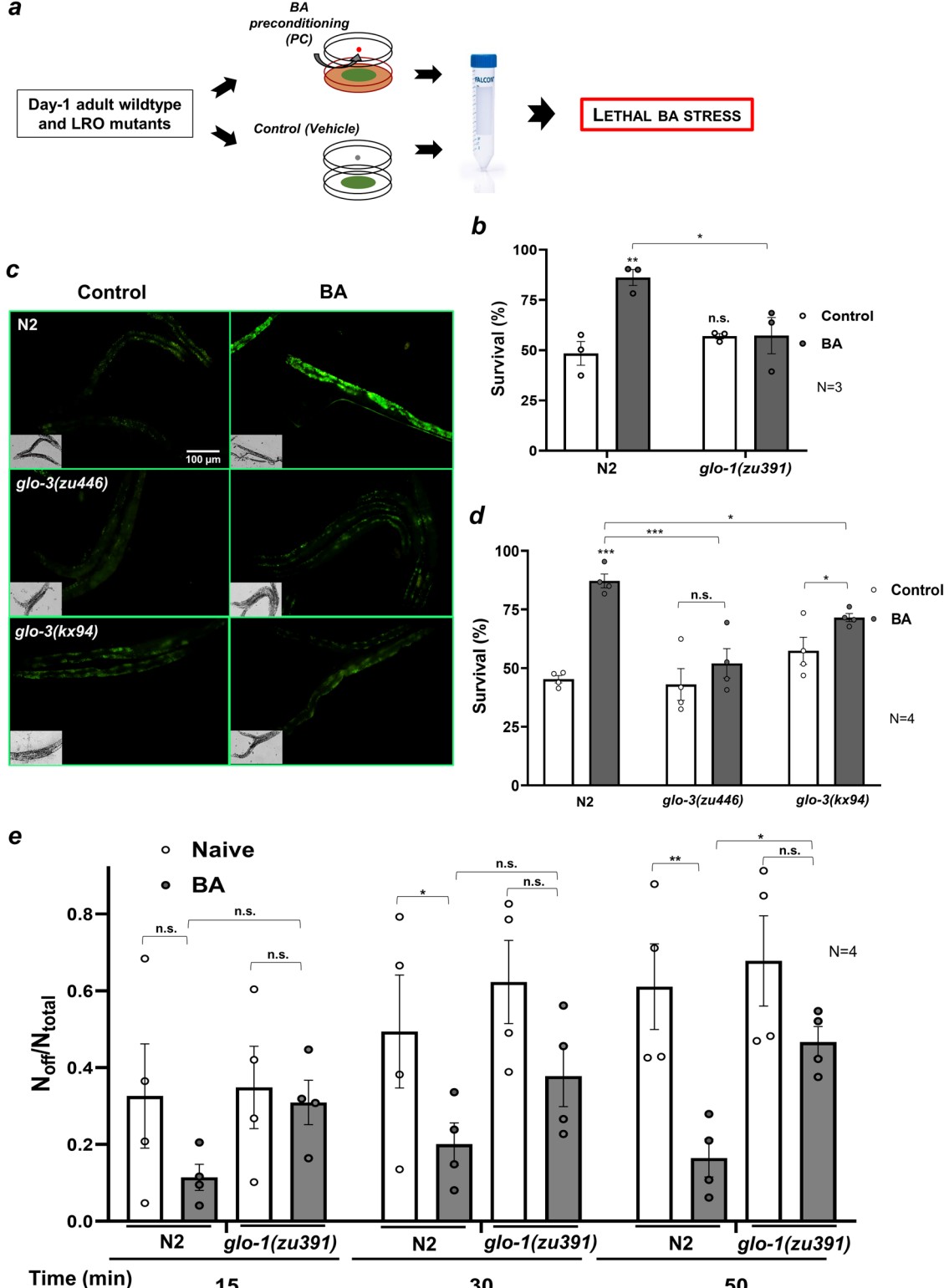

**Fig. 2 LRO biogenetic genes are necessary for physiological and behavioral tolerance to BA toxicity. a** Experimental setup to study the involvement of LROs in protection from lethal BA toxicity after a 4-h hormetic BA preconditioning. Wild-type and LRO-deficient mutant young adult worms were subjected to a hanging drop of hormetic BA or vehicle treatment, washed, and assayed for survival upon exposure to a lethal BA dose. **b** Survival assays of control and BA-preconditioned wild-type and *glo-1* worms exposed to a lethal dose of BA. **c** Fluorescence microscopy images of BA-exposed wild-type and two *glo-3* mutants. **d** Survival assays of control and BA PC wild-type and two *glo-3* mutants exposed to a lethal dose of BA. **e** BA-induced food avoidance of naive and BA PC wild-type and *glo-1* mutants. Data are expressed as mean ± SEM. N, number of independent experiments. *p*-values were obtained by two-way ANOVA with Fisher's LSD post hoc test. n.s., not significant; *$p < 0.05$; **$p < 0.01$; ***$p < 0.001$.

expected, lawn leaving of naive wild-type and *glo-1* strains were comparable, confirming they have a similar basal physiological response to this dose of BA (Fig. 2e). However, BA-preconditioned *glo-1* animals did not exhibit a significant reduction, contrasting a robust reduction of lawn avoidance of BA-preconditioned wild type (Fig. 2e). Thus, similarly to *daf-16* and *skn-1*[26], *glo-1* facilitates a cytoprotective response which enables animals to feed in an otherwise dangerous toxic milieu. As loss of *glo-1* ameliorates neuronal necrosis in a neurotoxicity model[33], at present it is not clear whether behavioral tolerance to BA is linked to the gut granule-related or a distinct, neuronal function of *glo-1*.

Consistent with our observations of the elevated autofluorescence and diminished physiological BA tolerance of *flu-1* mutants, AA treatment elevated blue autofluorescence only in wild-type, but not in *glo-1* worms (Supplementary Fig. 2d) and slightly, but non-significantly diminished avoidance of the BA-containing lawn in a *glo-1*-dependent manner (Supplementary Fig. 2e). These findings may suggest that the genetic or pharmacologic overload of intraorganellar AA might interfere with the metabolism or traffic of BA metabolite(s) which impairs the protection conferred by LROs against BA toxicity, which remains to be characterized.

**BA treatment enhances LRO transport capacity and stimulates the expression of LRO-specific genes.** In response to the increased burden, organelles initiate controlled compensatory processes (collectively called organellar stress responses) resulting in an increased functional capacity of the organelle[31]. The finding that LROs are required for a BA-elicited deposition of autofluorescent material and induction of organismal defense against BA toxicity prompted us to study how hormetic BA treatment affects the transport capacity of LROs. We monitored the kinetics of Nile Red filling. Control worms displayed an increased Nile Red accumulation at 120 min after an initial lag phase, suggesting a potential upregulation of dye transport. Furthermore, animals exposed to BA for 4 h prior to NR feeding exhibited an immediately increased red fluorescence which was significantly higher at all times than that of controls (Fig. 3a and Supplementary Fig. 3a). Notably, the elevated fluorescence was not due to the increased accumulation of hydrophobic material or defects in transport, as BA-treated animals exhibited a rapid release of Nile Red from their LROs which became identical to control worms within 30–60 min (Fig. 3b). These findings suggest an increased transport capacity of LROs in response to BA treatment.

Organellar adaptation responses are also characterized by the execution of specific transcriptional programs, such as the UPR^ER and UPR^MITO[31]. To address if the increased LRO capacity is accompanied by the expression of new genes, we measured the abundance of LRO-associated mRNAs in wild-type and *glo-1* worms after a 2-h and a 24-h BA treatment. First, we measured representatives of three relevant protein complexes involved in LRO biogenesis: *glo-1* and *glo-3* of the GLO-1, *apb-3* of the AP-3, and *vps-41* of the HOPS complex[7]. Whereas *glo-3* mRNA displayed enhanced, *glo-1*-independent expression in both time points, the *glo-1*, *apb-3,* and *vps-41* genes showed no significant induction (Fig. 3c and Supplementary Fig. 3b, e).

We also assayed three LRO-associated transporters. A predicted solute transporter *K09C4.5* was shown to be involved in LRO Nile Red accumulation in a *rict-1*-dependent manner[10], whereas the ABC transporters *pgp-2* and *mrp-4* possess roles in LRO biogenesis and maturation, respectively[14,34]. In response to BA treatment, we found a significant *glo-1*-dependent induction of *K09C4.5* mRNA at both 2 h and 24 h (Fig. 3d and Supplementary Fig. 3c). In addition, early and robust transcriptional induction of *pgp-2* with a weak *glo-1* dependency was observed, whereas the mRNA of *mrp-4* remained unchanged (Fig. 3d and Supplementary Fig. 3c). These

results together suggest that BA preconditioning engages a transcriptional response involving specific LRO-associated biogenesis and transport genes, and an increased LRO capacity and tolerance against BA toxicity. Altogether, *(i)* the expression of new genes supporting organelle function *(ii)* increased functional organelle capacity and *(iii)* elevated resistance to an organellar stressor may indicate a coordinated stress response.

**BA-induced LRO autofluorescence and gene expression require *daf-16*, *hsf-1*, and *pmk-1*.** BA exposure was shown to activate two cytosolic stress regulators, the DAF-16 and SKN-1 transcription factors, which, along with the *pmk-1* p38 MAP kinase ortholog were all required to survive BA toxicity[26]. We, therefore, asked whether these and other stress regulators might be involved in the organellar response of LROs. We observed that BA-induced autofluorescence was abolished not only in the *daf-16*, *skn-1,* and *pmk-1*, but also in the heat shock transcription factor *hsf-1* mutant backgrounds (Fig. 3e and Supplementary Fig. 3g). We note that nematode HSF-1 is constitutively localized in the nucleus which suggests that the BA-induced alterations reach the nucleus[35]. In contrast, the mitochondrial UPR regulator *atfs-1*, the *kgb-1* MAPK and the *kin*-1 protein kinase A catalytic subunit orthologs did not significantly affect the autofluorescence signal elevation (Supplementary Fig. 3d, g). Thus, an interplay of several stress regulators facilitates the accumulation of BA-related autofluorescent metabolites in LROs.

Next, we investigated whether the factors that regulate autofluorescence might be required for the transcriptional activation of LRO genes in response to BA. *pgp-2* induction was severely compromised in *daf-16* and *hsf-1* and was inhibited just below statistical significance in *pmk-1* mutants (Fig. 3f). *glo-3* induction upon BA was prevented by the loss of *pmk-1*, but already induced in the absence of *hsf-1* (Fig. 3g). Notably, the BA-induced expression of *K09C4.5* was not affected in the mutants tested, which indicates that its activation requires an unknown factor (Fig. 3h). We also note that the studied mutations did not alter the mRNA level of *glo-1* except a non-significant tendency to increase by the loss of *skn-1* function (Supplementary Fig. 3e). Moreover, the *skn-1* mutation did not affect the expression of LRO genes (Supplementary Fig. 3f). Altogether these findings suggest that metabolic processes and transcriptional responses in gut granules are orchestrated by specific stress pathway (*daf-16*, *hsf-1*, *pmk-1*) and unknown regulators through yet unidentified mechanisms.

**LROs are required for cross-tolerance to heat and oxidative stress.** Next, we asked whether LRO-mediated processes might play a role in tolerance to other environmental stresses. Both *hsf-1* and *daf-16* are important for survival under heat shock[36]. Therefore, we preconditioned worms with undiluted BA and measured their survival in response to heat shock. We found that BA preconditioning elevated the proportion of viable wild-type worms, while *glo-1* mutants exhibited a compromised basal heat stress tolerance compared to wild type, which was unaffected by BA PC (Fig. 4a). In contrast, preconditioning with DA or with AA neither improved thermotolerance in wild type, nor in *glo-1* mutants (Fig. 4b and Supplementary Fig. 4a), confirming a specific effect of BA through *glo-1*.

Then, we tested the role of LROs in the resistance against oxidative stress generated by paraquat, which generates superoxide through complex I of the mitochondrial respiratory chain[37]. We found that BA PC facilitated, while DA PC impaired the survival of wild-type worms (Fig. 4c, d). In agreement with its BA-related structure and activity[26], MS PC also promoted paraquat tolerance (Supplementary Fig. 4b). The loss of *glo-1* tended to reduce the basal resistance to paraquat (Fig. 4c, d, Supplementary Fig. 4b), but

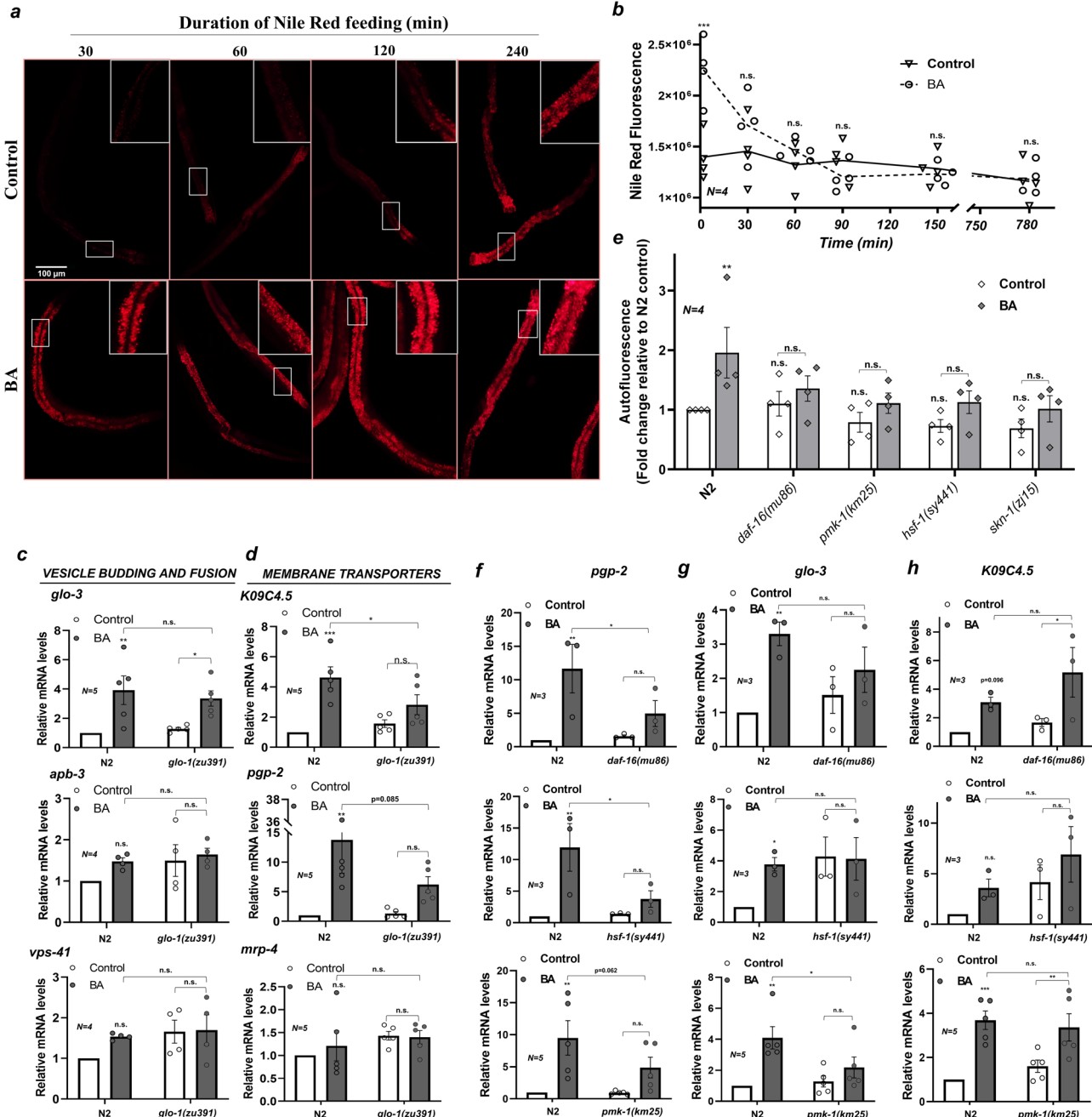

**Fig. 3 BA-induced, LRO-associated metabolic and transcriptional responses depend on diverse stress pathway regulators. a** Fluorescence microscopy images showing the effect of BA preconditioning on Nile Red accumulation. Worms were pretreated by vehicle or BA for 4 h then fed by Nile Red for various times as indicated. **b** Effect of BA preconditioning on Nile Red release. Vehicle- or BA-preconditioned worms were fed by Nile Red for 2 h, transferred to fresh plates and Nile Red fluorescence was quantified after the indicated times. **c** Relative mRNA abundance of genes involved in LRO-related vesicle budding and fusion of N2 and *glo-1* mutants after a 2-h BA treatment. **d** Relative mRNA abundance of LRO-related transmembrane transporters in N2 and *glo-1* mutants after a 2-h BA treatment. **e** BA-induced autofluorescence of N2 and various mutant nematodes. Worms were treated by BA for 4 h, then the green autofluorescence was quantified and expressed relative to the wild type control. **f–h** Relative abundance of *pgp-2* (**f**), *glo-3* (**g**), and *K09C4.5* (**h**) mRNAs in N2 and mutant worms after a 2-h BA treatment. Representative epifluorescence microscopy images were taken from two independent experiments. Data are expressed as mean ± SEM. N, number of independent experiments each in triplicates. *p*-values were obtained by two-way ANOVA with Fisher's LSD post hoc test. n.s., not significant; *$p < 0.05$; **$p < 0.01$; ***$p < 0.001$, ****$p < 0.0001$.

it did not reach statistical significance (N2 44% vs. *glo*-1 39% survival $p = 0.064$), suggesting that LROs might play a marginal role in basal resistance to oxidative stress. Importantly, both BA and MS augmented the survival to paraquat in a *glo-1*-dependent manner (Fig. 4c and Supplementary Fig. 4b). Altogether, these results indicate that *C. elegans* intestinal LROs contribute to basal heat stress tolerance as well as mediate the beneficial effects of

hormetic BA preconditioning to mount an adaptive organismal response to heat and oxidative stress.

**LRO-dependent activation of stress and innate immune responses.** We assumed that enhancing LRO function by BA preconditioning might promote cytoprotective stress and detoxification

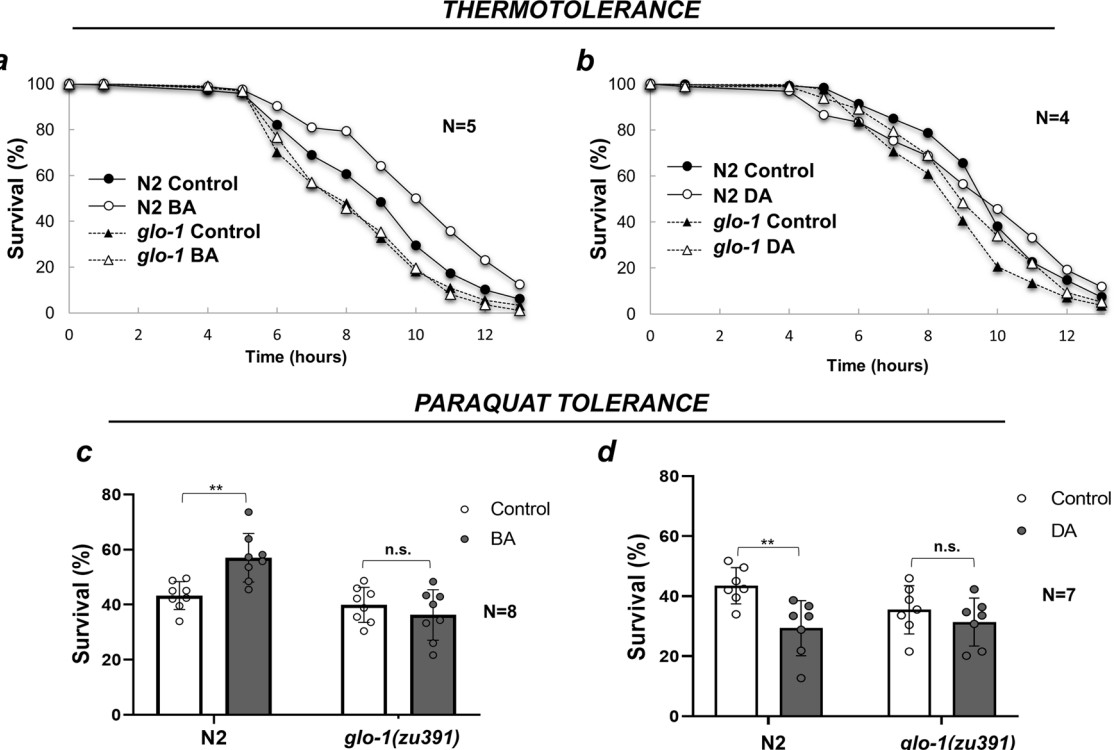

**Fig. 4 LROs mediate BA-induced cross-tolerance to heat shock and paraquat. a, b** Thermotolerance assays of N2 and *glo-1* animals at 35 °C, preconditioned (PC) by BA (**a**) or DA (**b**). BA PC significantly increases the survival of wild type ($p < 0.01$) but not that of *glo-1(zu391)* mutants. DA PC neither affects the survival of wild type nor that of *glo-1(zu391)* mutants. Overall data are plotted from 5 (BA) or 4 (DA) independent assays. Detailed statistics are given in Supplementary Table 1. **c, d** Paraquat toxicity assays of N2 and *glo-1* animals in 20 mg/ml PQ solution for 16 h, preconditioned by BA (**c**) or DA (**d**). Data are expressed as mean ± SEM. N, number of independent experiments each at least in triplicates. *p*-values were obtained by two-way ANOVA with Fisher's LSD post hoc test. n.s., not significant; \*\*$p < 0.01$.

responses. As BA forms protein adducts and causes oxidative stress[27], both might damage proteins, we determined the expression of molecular defense response genes to proteotoxic stress in N2 and *glo-1* nematodes after short- (2-h) and long-term (24-h) BA treatment. In wild-type worms, BA upregulated the *hsp-16.2* and *hsp-70* mRNAs representing the cytosolic heat shock response (HSR) especially at 24 h (Fig. 5a and Supplementary Fig. 5a). In contrast, BA did not affect the expression of *hsp-6* and *hsp-60* genes of the mitochondrial unfolded protein response (UPR-MITO) (Fig. 5b and Supplementary Fig. 5b), suggesting that the impact of BA does not involve mitochondria. Based on the observations that LROs accumulate BA or its derivatives, we reasoned that the loss of LROs might exaggerate the proteotoxic effect of BA, which in turn might activate the HSR. However, in *glo-1* mutants the HSR and the UPR-MITO were similar to wild type and were largely unaffected by BA at both time points (Fig. 5a, b and Supplementary Fig. 5a, b). We also tested whether the toxic load of BA in the absence of LROs might impair the induction of the HSR, by employing a similar, 4-h BA preconditioning as in the thermotolerance assay, followed by a 2-h heat stress. However, both wild-type and *glo-1* nematodes exhibited a similar, robust heat-induced HSR, which was unaffected by BA (Fig. 5c, d). Although a storage of BA or BA-related metabolites by LROs might contribute to the positive effects on heat stress tolerance, the above findings strongly suggest that in response to BA, *glo-1* is required to stimulate the cytosolic HSR.

Previously we observed that a 4-h BA treatment activated the promoters of two detoxification enzymes, the glutathione-S-transferase *gst-4* and the cytochrome P450 oxygenase *cyp-35B*[26]. We determined the mRNA level of these enzymes and found an early induction of *gst-4* and a delayed induction of *cyp-35B*, the latter required *glo-1* for its upregulation (Fig. 5e and

Supplementary Fig. 5c). These findings show an active involvement of *glo-1* in promoting some, but not other, BA-induced oxidative stress and detoxification defenses.

Emerging evidence shows that metazoans, including *C. elegans* survey core cellular processes, and the disturbance of these vital processes by microbial 'effectors' triggers widespread activation of cytoprotective stress, detoxification, and innate immune responses, called surveillance or effector-triggered immunity, as well as behavioral aversion[38-41]. To test the innate immune response, we selected the *irg-1*, *irg-5*, and *clec-60* genes involved in host defense against both Gram-negative and Gram-positive pathogens, such as *P. aeruginosa*, *E. faecalis*, and *S. aureus via* different antimicrobial pathways[38,42]. All genes were upregulated by BA at 24 h in the absence of pathogens (Fig. 5f and Supplementary Fig. 5d), suggesting that nematodes interpret BA exposure as a pathogen attack. Strikingly, all genes required *glo-1* for their BA-dependent induction (Fig. 5f). Taken together, our observations suggest that in response to BA, *glo-1* mediates a specific and widespread activation of stress, detoxification, and immune responses.

**BA-elicited defense responses require the *pgp-2* ABC transporter.** Similarly to mammals, ABC transporters function to protect *C. elegans* from xenobiotic and microbial toxins[43,44] and heavy metals[45]. In response to BA treatment we observed a robust early induction of *pgp-2*, but not *mrp-4* (Fig. 3d). Specifically, *pgp-2* is induced by the antihelminthic agent ivermectin in ivermectin-resistant worms[46] and the *pgp-2* loss-of-function mutant is more sensitive to ivermectin[47]. Therefore, we reasoned that the induction of *pgp-2* in response to BA treatment might be involved in the LRO-dependent protective response. Adult *mrp-4* worms possess a normal number of gut granules with no apparent phenotype[34],

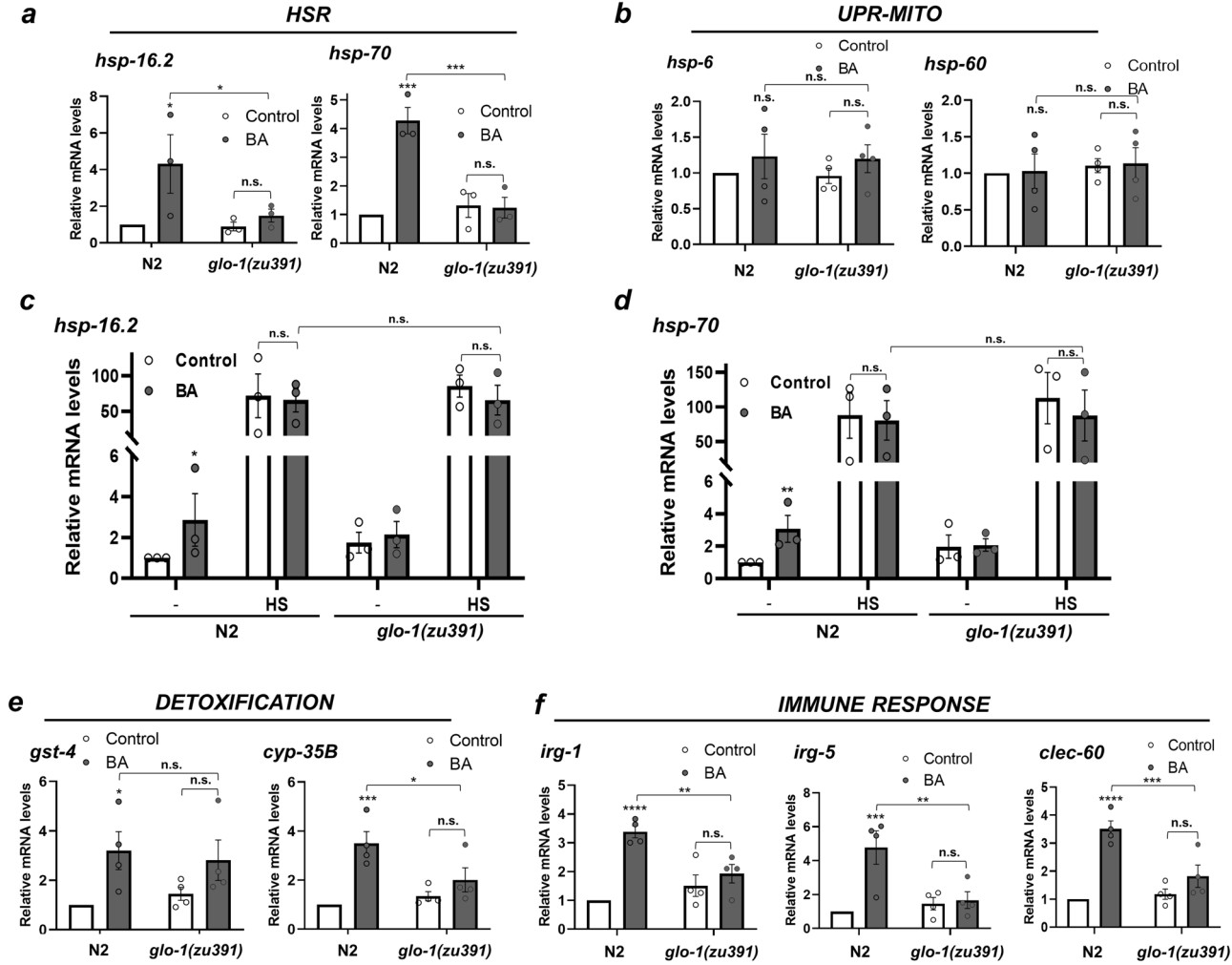

**Fig. 5 BA-induced, LRO-dependent expression of stress, detoxification, and immune responses. a, b** Relative mRNA abundance of the (**a**) cytoplasmic heat shock response (HSR) and the (**b**) mitochondrial unfolded protein response (UPR-MITO) in wild-type and *glo-1* nematodes after a 24-h BA treatment. **c, d** Relative abundance of *hsp-16.2* (**c**) and *hsp-70* (**d**) mRNAs after a 2-h heat stress at 31 °C, preceded by a 4-h vehicle or BA preconditioning treatment. **e, f** Relative mRNA abundance of detoxification enzymes (**e**) and antimicrobial effectors (**f**) after a 24-h BA treatment. $N = 4$ (or $N = 3$ in panels **c** and **d**, each in triplicates. *p*-values were obtained by two-way ANOVA with Fisher's LSD post hoc test. n.s., not significant; *$p < 0.05$; **$p < 0.01$; ***$p < 0.001$; ****$p < 0.0001$.

whereas the loss of *pgp-2* leads to the reduced number of non-acidified gut granules, suggesting a role for PGP-2 in a later stage of LRO biogenesis[14]. Using null alleles, we found that BA exposure led to wild type-like autofluorescence increase in *mrp-4* (Fig. 6a). However, instead of a granular autofluorescence, BA led to a faint cloudy signal in *pgp-2* mutants, probably because it could not reach the LROs (Fig. 6a). Likewise, although both mutants showed wild type-like basal tolerance to BA toxicity, loss of *mrp-4* did not affect the hormetic effect of BA preconditioning, whereas *pgp-2* mutation completely compromised it (Fig. 6b). In light of this, our results exclude a role for *mrp-4* and suggest that it is LRO biogenetic and/or transport function of PGP-2 which mediates the LRO localization of BA-induced autofluorescent material and protection from BA toxicity. To differentiate between these functions of *pgp-2*, we fed worms by *pgp-2* RNAi. RNAi treatment was employed from the L1 stage to silence *pgp-2* while preserving embryonic LRO biogenesis. *pgp-2* RNAi largely reduced the number of autofluorescent gut granules. These granules were able to accumulate Nile Red, which was modestly induced by BA treatment (Supplementary Fig. 6a, b). It is not known whether the functionality of the remaining granules is due to residual PGP-2 protein from the

embryonic period or to the suboptimal efficiency of the RNAi, which does not allow a clear conclusion on the functional role of *pgp-2* in BA-associated LRO transport.

We continued our studies with *pgp-2* by testing its impact on paraquat tolerance. *pgp-2* animals were unable to significantly increase their survival either upon 4-h or 24-h BA PC, demonstrating that *pgp-2* is required for cross-tolerance to paraquat-induced oxidative stress (Fig. 6c). Perhaps the robust early induction of *pgp-2* by BA (Fig. 3c) conferred protection against paraquat. We next asked whether *pgp-2* contributes to the elevated transcription of selected LRO-associated stress, detoxification, and antimicrobial genes. Apart from an elevation of *K09C4.5* levels in control *pgp-2* worms, the loss of *pgp-2* resulted in an mRNA expression pattern highly similar to that of *glo-1* worms (Fig. 6d–g): with the exception of *gst-4* whose mRNA level was independent of both *glo-1* and *pgp-2* at 24-hr, all the tested genes including *glo-3*, *hsp-16.2*, *hsp-70*, *cyp-35B*, *irg-1*, and *irg-5* were downregulated in both mutants (cf. the corresponding diagrams in Figs. 3, 5, and 6). Likewise, loss of *pgp-2* did not affect the induction of the HSR by heat stress, which was unchanged by BA preconditioning (Supplementary Fig. 6c, d), arguing against the

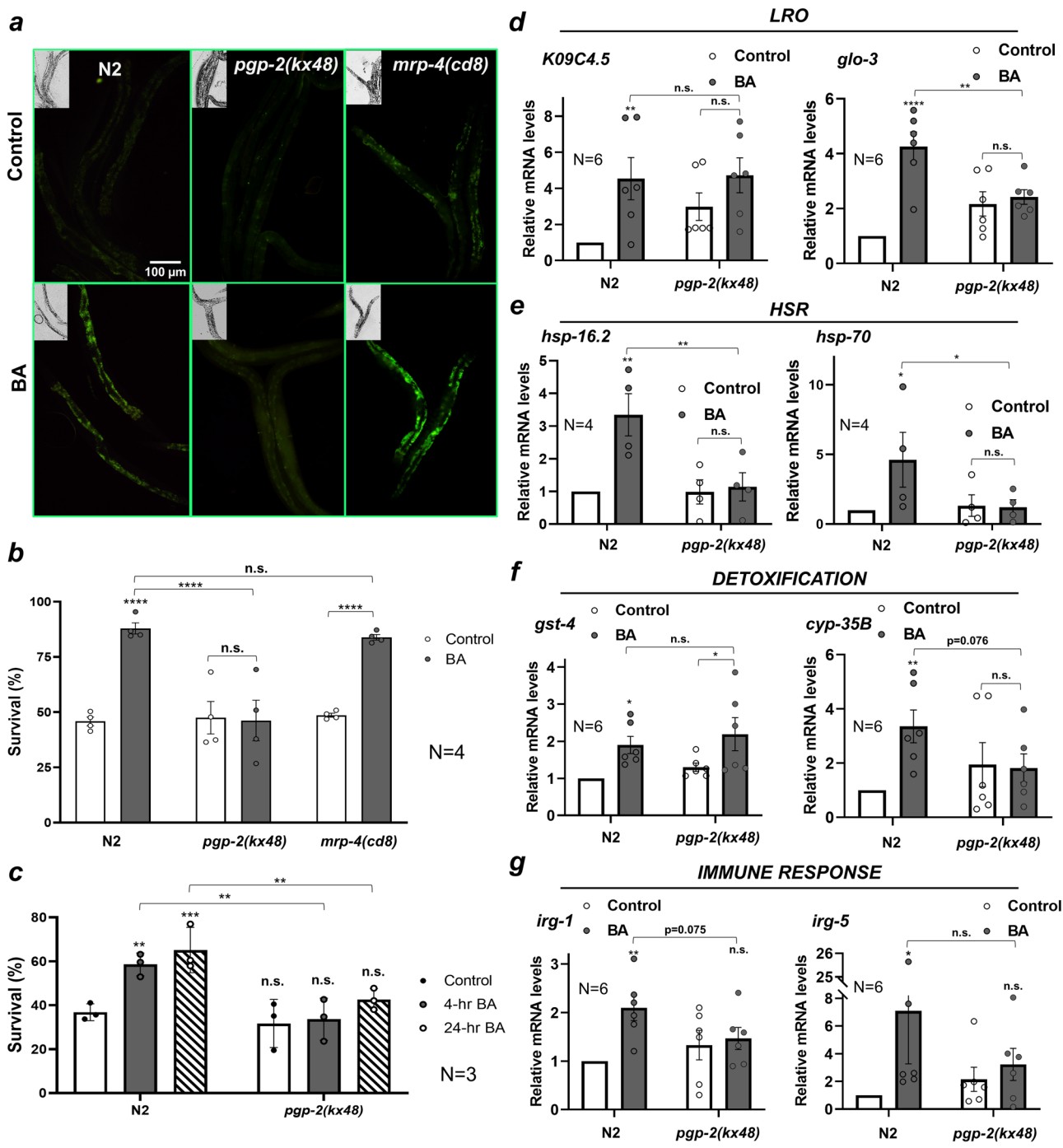

**Fig. 6 BA-elicited stress and immune responses depend on the *pgp-2* ABC transporter. a** Fluorescence microscopy images of BA-exposed wild-type *pgp-2* and *mrp-4* mutants. **b** Survival assays of control and BA PC wild-type, *pgp-2*, and *mrp-4* mutants exposed to a lethal dose of BA. **c** Paraquat tolerance assays of N2 and *pgp-2* animals preconditioned (PC) by 4-h or 24-h BA. **d–g** Relative mRNA abundance of LRO (**d**), heat shock response (**e**), detoxification (**f**), and immune-related genes (**g**) in wild-type and *pgp-2* animals after a 4-h (HSR) and 24-h BA treatment. N, number of independent experiments. *p*-values were obtained by two-way ANOVA with Fisher's LSD post hoc test. n.s., not significant; *$p < 0.05$; **$p < 0.01$; ***$p < 0.001$.

passive storage of toxic waste by LROs. Altogether, these results independently confirm those obtained with *glo-1* and further suggest the PGP-2 ABC transporter is required to mount an LRO-dependent systemic defense response with the potential to confer protection against various environmental assaults.

**LRO-dependent stress and immune responses require *daf-16* and *pmk-1*.** Then we set out to investigate the downstream mechanisms leading to the LRO-dependent expression of stress

and immune responses by using respective stress regulator mutants. Heat induction of *hsp-16.2* is mediated by a cooperative activation of DAF-16 and HSF-1, whereas that of *hsp-70* is mediated by HSF-1[36]. We found that the BA induction of *hsp-16.2* required *daf-16* and that of *hsp-70* required *pmk-1*, but neither of them was mediated by *hsf-1* (Fig. 7a, b, c). This result suggests that the LRO-dependent induction of these molecular chaperones does not use the canonic HSR, but is connected to oxidative stress and immune pathways. *daf-16* also mediated the

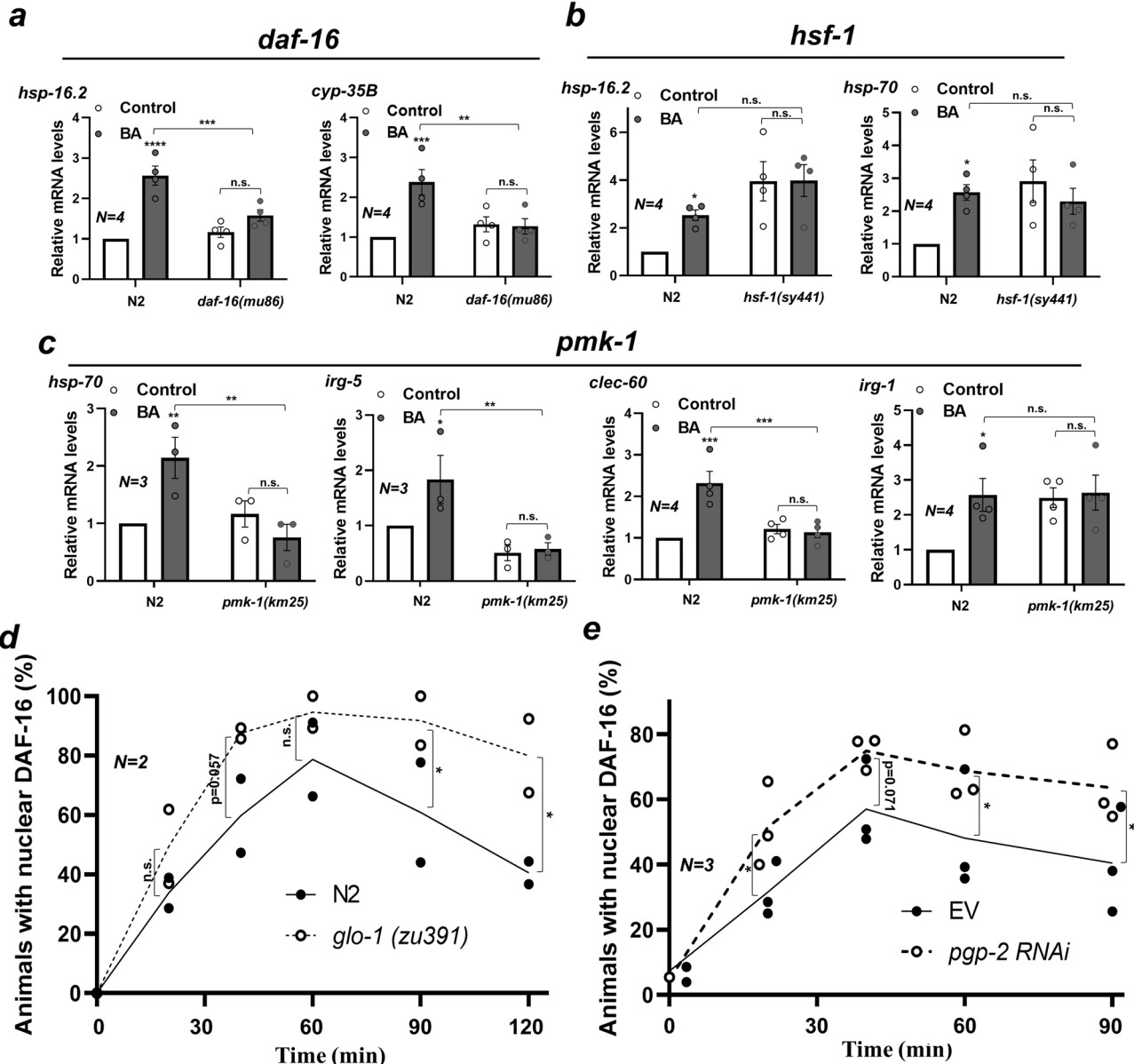

**Fig. 7 LRO-dependent stress and immune responses require *daf-16* and *pmk-1*. a, b** Relative abundance of BA-induced LRO-dependent mRNAs in wild-type and *daf-16* (**a**) and *hsf-1* (**b**) nematodes after a 24-h BA treatment. **c** Relative abundance of BA-induced LRO-dependent mRNAs in wild-type and *pmk-1* nematodes after a 24-h BA treatment. **d, e** Quantification of the nuclear localization of DAF-16::GFP at different time points during BA exposure in wild-type and *glo-1* nematodes (**d**) as well as in EV and *pgp-2* RNAi-fed wild-type nematodes (**e**). N, number of independent experiments. *p*-values of relative mRNA levels were obtained by two-way ANOVA with Fisher's LSD post hoc test, whereas *p*-values of DAF-16 localization were obtained by pairwise comparison of two-tailed Student *t*-test. n.s., not significant; *$p < 0.05$; **$p < 0.01$; ***$p < 0.001$.

BA-induced LRO-dependent activation of its target *cyp-35B*, whereas the detoxification regulator *skn-1* had no effect on *cyp-35B* expression (Fig. 7a and Supplementary Fig. 7a and ref. [26]).

DAF-16 was reported to translocate into the nucleus and activate the *cyp-35B* promoter in response to BA exposure[26]. Based on the lost *hsp-16.2* and *cyp-35B* induction in *glo-1* and *pgp-2* LRO mutants (Figs. 5a, e and 6e, f), we hypothesized that LROs might interfere with the activation of DAF-16-mediated transcription. Indeed, in response to BA, DAF-16::GFP exhibited a higher and more persistent nuclear localization in *glo-1* mutants and *pgp-2* RNAi-fed worms (Fig. 7d, e). This finding indicates that in the absence of LROs, compensatory nuclear retention of DAF-16 occurs as a result of its deficient transcriptional activity. The inability of DAF-16 to induce its targets in LRO mutants together with its nuclear retention strongly suggests that LRO-

dependent signals are indispensable for DAF-16 transactivation in response to BA stress. In contrast, BA-induced SKN-1::GFP nuclear translocation was unaltered by *pgp-2* RNAi (Supplementary Fig. 7b), so was the induction of the *skn-1* target gene *gst-4* in LRO mutants (Figs. 5e and 6f; ref. [26]).

As the stress and immune regulator PMK-1 is involved in BA toxicity[26], we tested its potential involvement in the LRO-dependent expression of antimicrobial genes. Indeed, the BA-induced expression of the known *pmk-1* target *clec-60*[48] was disrupted, whereas that of the *zip-2* target *irg-1*[38] was unaffected in *pmk-1* mutants (Fig. 7c). *irg-1* was neither affected by *skn-1* mutation (Supplementary Fig. 7c). The PA14-induced expression of *irg-5* is known to require both *pmk-1* and the nuclear receptor *nhr-86*[49]. Consistent with this, we found that BA-induced expression of *irg-5* was compromised in both *pmk-1* and *nhr-86*

mutants (Fig. 7c and Supplementary Fig. 7d). Moreover, *pmk-1*, but not *atfs-1* mediated BA PC-induced paraquat tolerance (Supplementary Fig. 7c). These findings together with the *pmk-1*-dependent induction of *hsp-70* suggest that in response to a toxic insult by the volatile BA, LROs and PMK-1 cooperate in the elicitation of stress and antimicrobial defenses.

**LROs are required for resistance against *P. aeruginosa* infection.** *The* robust antimicrobial defense system of *C. elegans* is primarily regulated at the transcriptional level[41]. Because all three antimicrobial effectors tested were induced by BA in an LRO-dependent manner, we set out to study whether BA PC affects bacterial pathogen resistance. We chose the *Pseudomonas aeruginosa* PA14 strain keeping in mind the strong virulence and serious threat of various *P. aeruginosa* strains as multiresistant opportunistic pathogens[50]. Major pathogenetic factors in *P. aeruginosa* infection are diffusible bacterial toxins, primarily phenazines, and involve oxidative stress in the host[44,51]. Indeed, we observed an induction of *cyp-35B* and *gst-4* detoxification genes in response to PA14 infection, which appeared to partially depend on *glo-1* (Supplementary Fig. 8a, b). To assess the resistance against pathogen toxic burden, we employed the fast killing protocol developed in the Ausubel lab, where killing predominantly occurs through toxins[44]. We note that fast killing of day-1 adults occurs at a slower rate than that of L4 larvae[52]. As expected, BA preconditioning significantly elevated survival at each time point tested, resulting in a 19% increase in mean survival (Fig. 8a–d). Both untreated *glo-1* and *pgp-2* mutants exhibited diminished mean survival, which in the case of *pgp-2* was more pronounced and highly significant (Fig. 8b, d). Moreover, neither of the two LRO mutants' pathogen resistance was affected by BA PC (Fig. 8a–d).

We then asked whether LROs are involved in the pathogen response elicited by *P. aeruginosa*. Hence, we measured the expression of the antimicrobial effector genes that were induced by BA PC (see Figs. 5e and 6f) after a 16-h PA14 infection in wild-type and LRO mutant animals. Wild-type worms upregulated *irg-1* and *irg-5*, but not *clec-60* upon PA14 exposure (Fig. 8e, f). In contrast to the BA-elicited induction of *irg-1*, *glo-1*, and *pgp-2* were not required for that by PA14 infection, suggesting different LRO-dependent and LRO-independent mechanisms leading to *irg-1* upregulation. However, both *glo-1* and *pgp-2* were indispensable for the PA14-induced expression of *irg-5*, demonstrating that the pathogen-induced response involving *irg-5* requires functional LROs (Fig. 8e, f). It was demonstrated that *irg-5* knock-down per se compromises PA14 resistance[49], suggesting it might be an important mediator of the anti-pathogen action of LROs. Interestingly, whereas the loss of *glo-1* did not affect the expression of the two other genes, the absence of *pgp-2* robustly stimulated *irg-1* and induced *clec-60* mRNA levels in response to PA14 infection (Fig. 8e, f).

Pathogen attack-induced host damage elicits different organellar stress responses such as the endoplasmic reticulum and mitochondrial UPR, which in turn besides promoting cellular repair contribute to the activation of antimicrobial effectors[53,54]. To test whether PA14 infection induces the LRO stress response, we assessed the expression of the LRO genes shown to be activated by BA (Fig. 3). In wild-type nematodes, *glo-3* remained unchanged, while *K09C4.5*, *pgp-2* and surprisingly, *glo-1* were upregulated by a 16-h PA14 exposure (Fig. 8g, h). Apart from an increased basal expression, PA14 did not affect the level of *K09C4.5* and *pgp-2* mRNAs in *glo-1* mutants. Similar to the observations in the case of the antimicrobial genes, the absence of *pgp-2* further enhanced *K09C4.5*, induced *glo-3* mRNA levels, and inhibited *glo-1* upregulation in response to PA14 infection. The

compensatory expression of several LRO and antimicrobial genes and the prevention of *glo-1* and *irg-5* induction might explain the detrimental effects of PA14 infection if *pgp-2* was lost (Fig. 8f, g), suggesting a critical role of PGP-2 in pathogen defense.

We reasoned that the PA14-induced LRO stress response might be a consequence of an increased LRO demand during infection. To test this possibility, we took use of the fact that *Pseudomonas* virulence factors are blue-green colored fluorescent pigments, namely phenazines (such as pyocyanin) and siderophore pyoverdines[55,56]. We exposed wild-type and LRO-deficient worms to PA14 for four h and examined them under a fluorescence microscope using an IX3-FBVWXL 460 nm long-pass filter. On *E. coli* OP50, wild-type animals exhibited a usual green granular autofluorescence emitted by intestinal LROs (Fig. 8i). On PA14, gut granules turned into brilliant yellow-gold, and little fluorescent material was detected in the intestinal lumen. This pattern was entirely missing in both *glo-1* and *pgp-2* mutants. Instead, we observed a strong blue-green fluorescence in the gut lumen (Fig. 8i), perhaps due to the lack of transport and metabolism of PA14 pigments into yellow-gold products in the absence of LROs. Altogether, these findings demonstrate that in response to PA14 exposure, worms accumulate distinct, yet unknown metabolites in intestinal LROs, activating an LRO stress response which in turn promotes pathogen defense. This process beyond the GLO-1 small GTPase critically involves the PGP-2 ABC transporter and can also be activated by the LRO inducer BA.

**Discussion**
The current study shows that the undiluted toxic volatile benzaldehyde (BA) elicits the deposition and/or modification of autofluorescent material(s), the expression of LRO-associated genes, and the increased functional capacity of *C. elegans* gut granules. In turn, this LRO response is indispensable for the elevation of organismal tolerance to BA toxicity, heat, oxidative and pathogen stresses *via* the upregulation of various stress, detoxification, and innate immune responses. PA14 infection also results in the deposition of autofluorescent material and in the induction of LRO genes. These processes depend on the Rab32/38 GTPase ortholog *glo-1* and the ABC transporter *pgp-2* and require predominantly the *daf-16* and *pmk-1* stress and immune regulators. Our findings imply a hitherto unidentified function of LROs connected to conserved stress, detoxification, and innate immune pathways which promotes systemic defenses against environmental noxae.

In response to undiluted BA, gut granules rapidly increase their green and blue autofluorescence (Fig. 1 and Supplementary Fig. 2). A similar, slower increase in LRO autofluorescence occurs during aging, which might originate from anthranilic acid (AA) glucoside[21]. However, the autofluorescence observed upon BA exposure was unrelated to anthranilate formation, and a similar autofluorescent signal was detected upon methyl-salicylate (MS) exposure (Supplementary Fig. 1). Strikingly, PA14 infection also robustly increased gut granule fluorescence (Fig. 8). All BA, MS, AA, and *Pseudomonas* phenazines possess substituted aromatic benzene rings which are known to fluoresce in acidic milieu[57]. Perhaps the aromatic compounds themselves are transported (and metabolized) in LROs which elevates autofluorescence. Alternatively, these molecules elicit cellular responses leading to the chemical modification of other molecules either resident or transported into the LROs. Further research is needed to identify the compounds and the mechanisms leading to increased LRO autofluorescence.

We note that all the above compounds exhibit toxicity[27,55,58,59]. An important step in cytoprotection is the detoxification and sequestration of potentially dangerous lipophilic compounds by

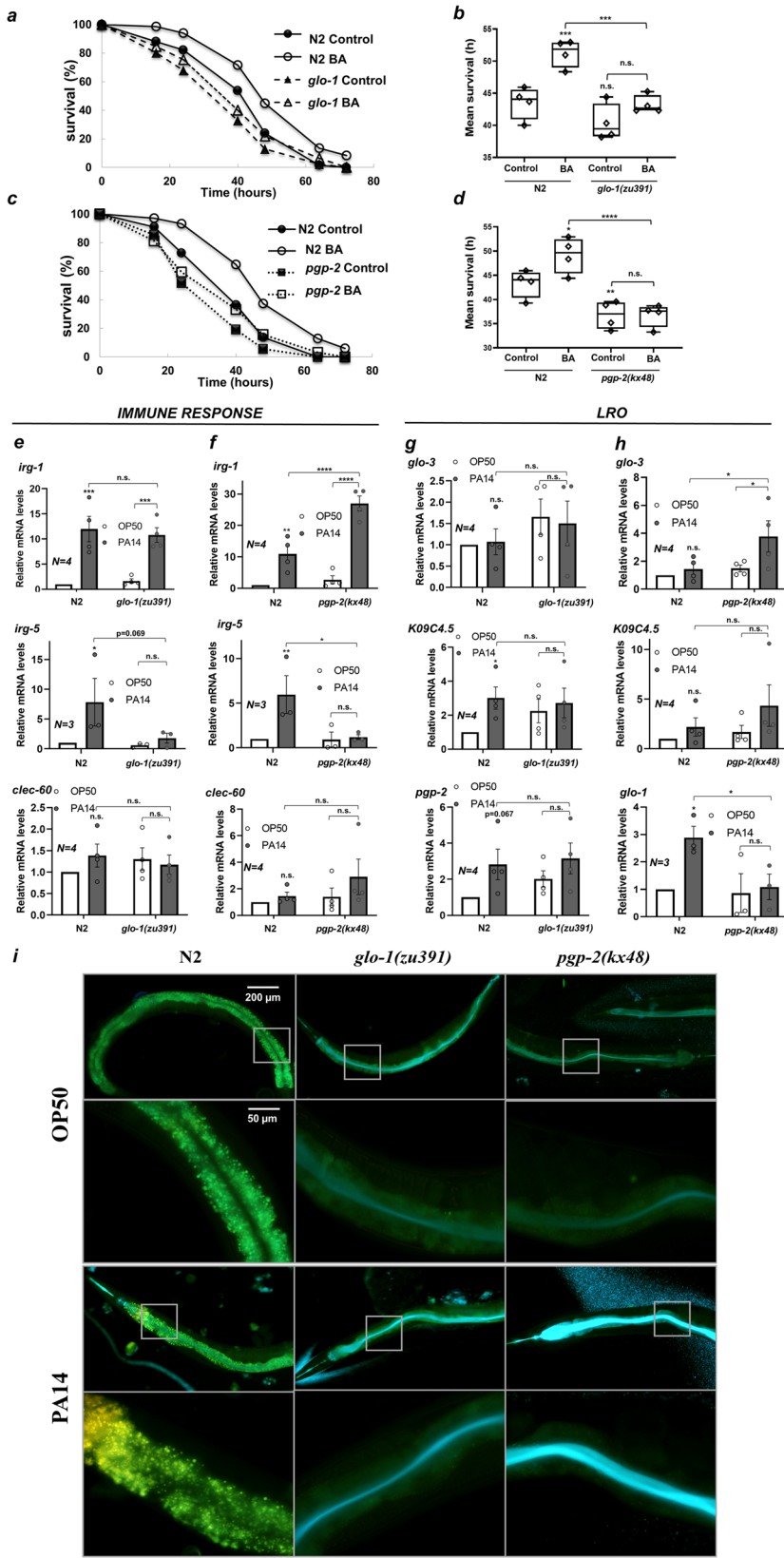

enzymes and transporters in membrane-bound compartments, such as the ER. For LROs, the deposition of AA glucoside conjugates has been demonstrated[21]. The proposal on the deposition of AA (and perhaps other compounds) as intrinsically formed toxic waste is supported by the lifespan extension in *kynu-1* mutants that do not produce AA[60] as well as reduced lifespan of LRO mutants[24].

Hence, the vesicular accumulation of metabolite(s) upon exposure to exogenous toxic compounds, such as BA, MS, and microbial toxins might be a similar protective measure that represents a previously unrecognized function of *C. elegans* LROs. The finding that BA-induced autofluorescence requires *daf-16*, *pmk-1*, *hsf-1*, and *skn-1* (Fig. 3) suggests extensive connections with other stress

**Fig. 8 LROs are required for resistance against *P. aeruginosa* infection. a, b** A representative PA14 killing assay (**a**) and comparison of mean survivals (**a**) of control and BA PC wild-type and *glo-1* animals. **c, d** A representative PA14 killing assay (**c**) and comparison of mean survivals (**d**) of control and BA PC wild-type and *pgp-2* animals. Killing assays were repeated four times and compared by the log-rank (Mantel-Cox) test. Data are expressed as mean ± SEM. Detailed statistics are given in Supplementary Table 2. **e, f** Relative mRNA abundance of antimicrobial effector genes in wild-type, *glo-1* (**e**), and *pgp-2* (**f**) animals after a 16-h PA14 exposure. **g, h** Relative mRNA abundance of LRO-specific genes in wild-type, *glo-1* (**g**), and *pgp-2* (**h**) animals after a 16-h PA14 exposure. N, number of independent experiments. *p*-values were obtained by two-way ANOVA with Fisher's LSD post hoc test. n.s., not significant; **p* < 0.05; ***p* < 0.01; ****p* < 0.001. **i** Fluorescence microscopy images of wild-type, *glo-1* and *pgp-2* mutants after a 16-h exposure to PA14. Scale bar for PA14 is the same as that for OP50. Every second row shows the magnified inset. Please note the gold granular fluorescent signal in the intestine of wild-type animals, and the blue-green signal in the intestinal lumen of the mutants after PA14 exposure. Representative images of three independent experiments with similar results.

pathways all implicated in the repair of toxic insults (such as oxidative and electrophilic agents and proteotoxicity) as well as in innate immunity[61–64].

We observed that hormetic preconditioning with BA conferred *glo-1*- and *glo-3*-dependent organismal protection from a subsequent lethal BA exposure (Fig. 2b, d). The induced toxic BA tolerance together with *(i)* increased LRO traffic/deposition of metabolites *(ii)* increased expression of LRO-specific mRNAs including biogenesis genes and membrane transporters and *(iii)* enhanced LRO Nile Red staining (Figs. 1 and 3) indicate an efficient cytoprotective stress response involving gut granules. A similar role of LROs was shown in zinc toxicity with an increased zinc-binding capacity, an altered, bilobed shape of the LROs, and specific protection against zinc, but not cadmium toxicity, which required the CDF-2 cation transporter[23]. Although we did not investigate the detailed morphology of LROs, our study extends the protective role of LROs against non-ionic compounds and shows a transcriptional adaptive response. Besides, the reduced lawn avoidance mediated by *glo-1* in BA-preconditioned worms (Fig. 2) extends previous findings on the impact of cytoprotective responses in behavioral decisions[26]. To find potential mediator(s) responsible for upregulation of LRO mRNAs, we screened large-scale transcriptomic data for the following stress-responsive regulators: the lysosome biogenetic HLH-30[65], the intestinal ELT-2[66], the aryl-hydrocarbon receptor AHR-1[67], the PMK-1-controlled oxidative and immune regulator ATF-7[68], the heat shock factor HSF-1[69], the major IIS effector DAF-16[70] and the xenobiotic regulator SKN-1[71], the latter two were reported to be activated by BA and MS[26]. Although we did not find evidence for a considerable regulation of LRO genes by these factors in response to their canonic activators, we found that BA-induced expression of *pgp-2* required primarily *daf-16* and *hsf-1*, and that of *glo-3* required *pmk-1*. It remains to be tested whether *daf-16* and *pmk-1* might co-regulate these and other BA-induced LRO genes. However, the complete *daf-16*- and *pmk-1*-independence of *K09C4.5* induction argues against an exclusive role of these factors (Fig. 3). Thus, our initial findings on selected LRO genes already suggest that the LRO compartment might possess an intricate regulation through known and unknown signaling mechanism(s). Harnessing benzaldehyde's stimulatory effect might foster the systematic exploration of the molecular mechanisms of the LRO-associated stress response.

Our findings on the reduced thermotolerance of 1-day-old *glo-1* mutants confirmed earlier results obtained on 5-day-old LRO mutants[24]. Moreover, we found that the induction of LRO response by BA-elicited cross-tolerance to heat and oxidative stress (Fig. 4 and Supplementary Fig. 4). On the contrary, the genetic and pharmacologic elevation of AA levels, respectively, impaired all the increase of BA-induced autofluorescence, tolerance to lethal BA and heat stress (Supplementary Figs. 2 and 4). Although the reason for the negative effects of AA is unknown, we speculate that AA or its derivative might inhibit some protective functioning of the LROs. Such a putative LRO-inhibitory action of AA might be the subject of future studies.

The opposing effect of AA and BA/MS on stress tolerance and the apparent health and wild-type-like basal stress tolerance of young LRO mutants suggested it is not merely a deposition of toxic metabolites in gut granules, but their active involvement in an organismal response to stress. Indeed, we showed the stimulation of carefully selected marker genes of specific stress, detoxification, and innate immune responses (Fig. 5) and food avoidance by BA[26]. It may well be that BA and other toxic agents, such as those of PA14 induce a more comprehensive stress response, which, together with the underlying LRO-dependent mechanisms are the subject of a future system-wide research project. The modest upregulation may prime widespread systemic defenses to confer organismal protection against different stresses. These findings are consistent with the effector-triggered immunity elicited by the disruption of host physiology by pathogens or toxins[38–41]. The requirement of *glo-1* and the ABC transporter *pgp-2* in the elicitation of the protective responses as well as their role in PA14 infection (Figs. 5, 6, and 8) suggest LRO-associated mechanisms as a novel cellular surveillance program, which helps prepare host defense against various, including pathogen stresses. Whether the deposition of toxic metabolites in gut granules and the LRO-dependent elicitation of stress responses are separate or intertwined processes, as well as the extent of their contributions towards organismal protection, are open questions. The intestinal location of LROs gains relevance for the entire organism, as the non-professional immune cells of the intestinal epithelium are a major barrier in contact with the environment. BA is a nutrient-signaling bacterial metabolite that is attractive at lower concentrations. However, a higher BA dose reaching the gut might serve as a sign of live propagating microorganisms in the interior, which is known to activate effector-triggered immunity[40]. These findings would also provide a plausible explanation for the diminished lifespan of LRO mutants[24] and are consistent with the diminished lifespan and stress tolerance of other cytoprotective regulator mutants such as HSF-1, DAF-16, and SKN-1[64,72].

The exposure of nematodes to BA and MS was previously shown to induce DAF-16 and SKN-1 nuclear translocation and expression of DAF-16-dependent *cyp-35B* and SKN-1-dependent *gst-4* reporters, respectively[26]. Here, we observed that *glo-1* and *pgp-2* were required for BA-induced elevation of the DAF-16 target *cyp-35B* and of the obligate HSF-1 targets *hsp-70* and *hsp-16.2*, the latter requires both *hsf-1* and *daf-16* for heat-induced expression[36] (Figs. 5 and 6). Our further experiments confirmed a *daf-16*-dependent activation of *cyp-35B* and *hsp-16.2* and demonstrated that in the absence of LROs, DAF-16::GFP exhibited increased and more persistent nuclear translocation, suggesting that yet unknown LRO-specific signals are indispensable for DAF-16 to activate the expression of its targets in response to BA. The BA-induction of *hsp*-s did not require *hsf-1*. Instead, *hsp-70* induction was dependent on *pmk-1*. Finally, SKN-1 and LRO-associated pathways appear to operate independently, whereas the mitochondrial UPR and its regulator *atfs-1* are not involved in the BA stress response

(Figs. 5 and 7; Supplementary Fig. 7), suggesting specific crosstalks between LROs and some stress pathways.

Interestingly, both *daf-16* and *pmk-1* are involved in the expression of some LRO-associated genes, the accumulation of autofluorescent metabolites in LROs, and the LRO-dependent expression of stress and antimicrobial genes, which suggests a cooperative action and a role for LROs in feed-forward regulation of these factors. Whether LRO-derived signals play a permissive or an active role in this regulation, remains to be investigated. Notably, both the *pmk-1* and *daf-16* pathways are required for effective immunity[61,62]. In this regard, a relevant observation is the *glo-1*-dependent expression of several intestinal antimicrobial genes (Fig. 5) known to protect against extracellular pathogens such as *P. aeruginosa* PA14[38]. Consistent with this, *glo-1* is required for the increased resistance against PA14 by BA preconditioning (Fig. 8). An established role of mammalian Rab32 in host defense against intracellular pathogens involves its GTPase activity and trafficking function[73–76]. Although a similar role for GLO-1 might be plausible against the intracellular worm parasites *Microsporidia*[77,78], in our experiments *glo-1* appears to operate *via* another mechanism. In agreement with this idea, *glo-1* mediates the induction of the PMK-1 and NHR-86 target *irg-5* upon PA14 infection (Fig. 8 and Supplementary Fig. 8), which suggests a crosstalk between LRO-s and this nuclear hormone receptor[49]. A recent systematic study reported the *glo-1*-dependent synthesis of complex ascaroside and glucoside conjugates also utilizing AA and benzoic acid derivatives[25]. We speculate that these molecules might serve as signals to elicit the activation of effector-triggered immunity, perhaps through some of the 284-membered worm nuclear hormone receptor family. However, testing this assumption awaits further systematic experiments.

The replication of the diminished pathogen resistance and abolished *irg-5* induction in a *pgp-2* background (Fig. 8), where GLO-1 is active, further confirms that not merely the GLO-1 GTPase/vesicular trafficking activity, but rather an LRO-specific process is responsible for the pathogen response. Notably, *pgp-2*, but not *glo-1* mutants retain the majority of LRO zinc storage capacity[23], which shows a selective requirement of the *pgp-2* ABC transporter in BA-induced pathogen resistance. The findings that loss of *pgp-2* compromises pathogen resistance despite activating both antimicrobial (*irg-1*) and LRO-specific (*glo-3* and *K09C4.5*) genes upon PA14 infection (Fig. 8) argue for a critical role of *pgp-2* in innate immunity. Such a role for *pgp-2* and for LROs is reinforced by the fact that *glo-1* is induced by PA14 in a *pgp-2*-depedent manner. Our temporal gene silencing experiments using *pgp-2* RNAi does not allow a clear conclusion on whether PGP-2 affects LRO biology *via* its specific ABC transporter function.

Taken together, our study reveals a previously unrecognized cytoprotective function of *C. elegans* gut granules, identifying an LRO inducer and a potential organellar stress response in response to toxic and pathogen insults. Besides, the findings presented here illuminate a role for LROs in mounting systemic stress, detoxification, and antimicrobial defenses, which requires a co-operation between ancient and highly conserved processes such as compartmentalization in the endolysosomal system, an ABC membrane transporter and transcriptional stress and innate immune pathways. Thus, the self-protective role of LROs emerged early in evolution.

## Methods

**Materials**. The reagents benzaldehyde, diacetyl, methyl-salicylate, Nile Red, anthranilic acid, and paraquat-dichloride hydrate were obtained from Sigma Aldrich. For quantitative gene expression analysis, primers were purchased from Sigma Aldrich. The GeneJet RNA purification kit, Proteinase K, Maxima RT enzyme, Ribolock RNase inhibitor, RT buffer, and dNTP were acquired from Thermo Scientific. Random Hexamer was from Invitrogen.

***C. elegans* strains and maintenance**. N2 Bristol (wild type), JJ1271 [*glo-1(zu391)* X.], GH383 [*glo-3(zu446)* X.], GH403 [*glo-3(kx94)* X.], GH378 [*pgp-2(kx48)* I.], GH534 [*mrp-4(cd8)* X.], CB1003 [*kynu-1(e1003)* X.], CB1002 [*flu-1(e1002)* V.], KU25 [*pmk-1(km25)* IV.], CF1038 [*daf-16(mu86)* I.], QV225 [*skn-1(zj15)* IV.], PS3551 [*hsf-1(sy441)* I.], KU21 [*kgb-1(km21)* IV.], QC115 [*atfs-1(et15)* V.], CZ3086 [*kin-1(ok338)/unc-54(r293)* I.], VL491 [*nhr-86(tm2590)* V.], TJ356 [daf-16p::daf-16a/b::GFP + *rol-6(su1006)* II.] LD1 [skn-1b/c::GFP + *rol-6(su1006) II*.] were provided by the Caenorhabditis Genetics Center. The strain SCS30 [TJ356 x JJ1271 (glo-1 (zu391) X.)] was generated by mating male *glo-1(zu391)* worms with TJ356 hermaphrodite. F2 offspring generation of the transgenic line was monitored for the lack of gut granules along with visible DAF-16::GFP signal. Strains were maintained on *E. coli* OP50 at 20 °C according to[79]. Specifically, synchronized populations were raised by allowing 10–15 adults to lay eggs for 4 h in case of behavioral assays, or by harvesting eggs from heterogeneous, well-fed populations in case of toxicity assays, quantitative PCR, or fluorescence microscopy. Day 1 adults were used at 20 °C for all experiments, except for thermotolerance assays.

**RNA interference**. The following HT115(DE3) *E. coli* dsRNA-producing strain was used in the study: *pgp-2* (Greg Hermann, Department of Biology, Lewis & Clark College, Portland, OR, USA). RNAi treatments were performed according to ref.[80]. RNAi-feeding clones were grown overnight in LB medium containing 100 μg/ml ampicillin. Briefly, synchronized L1 larvae were placed on plates seeded with *E. coli* HT115 strains harboring the L4440 empty vector (EV) control and *pgp-2* RNAi vector, respectively.

**Odorant treatments**. Treatments of synchronous young adult populations with the odorants benzaldehyde, methyl-salicylate, and diacetyl were introduced by our laboratory[26], using a hanging drop of 1 μl undiluted, 100% (v/v) odorant on the lid of 6 cm NGM-plates seeded with OP50 and sealed with Parafilm for the times indicated at the respective Figures. Preconditioning before lethal stresses or food avoidance assay was for 4 h, using 150–200 worms in a 6 cm NGM plate with OP50. Anthranilate treatment was conducted by dissolving from a 55 mM stock (in PBS) in warm liquid NGM agar at a final concentration of 0.55 mM, using 6 cm plates. Anthranilate treatment was for overnight using 150–200 worms per plate.

**Nile Red feeding assay**. Feeding of Nile Red was carried out by freshly diluting from a 500 μg/ml stock solution in acetone into a final concentration of 50 ng/ml in M9, dropping 100 μl onto the surface of the bacterial lawn, and being allowed to dry. Animals were kept on NR-supplemented plates for the times indicated in the respective Figure legends. Prior to imaging, animals were transferred to Nile Red-free OP50 seeded NGM plates for 20 min.

**Odorant toxicity assay**. Survival upon exposure to lethal doses of the odorants benzaldehyde and diacetyl was determined according to[26]. In detail, approximately 25–40 synchronized young adults in triplicates were picked onto 3 cm NGM plates seeded with OP50 and exposed to 2 μl undiluted BA hanging drop for 3 h. Worms were scored 14 h later by tapping with a platinum worm pick. Animals that crawled off the agar surface were censored.

**Food avoidance assay**. Aversion was measured in triplicates according to ref. [26]. Specifically, 50–80 synchronized young adults were used on 6 cm NGM plates with a small central lawn of OP50

containing 1 μl BA or 4 μl DA on a piece of parafilm in the middle. At the indicated times, the number of worms on and off the lawn was counted. Aversion index was calculated as $N_{off}/N_{total}$.

**Thermotolerance measurement**. Thermotolerance was assayed at 35 °C in triplicates, by picking 25–40 synchronized young adults onto OP50 seeded 6 cm NGM plate. After the restoration of motility, viability was determined by the absence of movement in response to gentle tapping of the plate. Animals that crawled off the agar were censored.

**Paraquat tolerance**. Paraquat (PQ) tolerance was performed according to ref. [81], in quadruplicates. In detail, 10–15 synchronized young adults per well were used a 96-well flat-bottom plates, containing 50 μl 20 mg/ml PQ solution in M9 for 16 h. Survival was analyzed by scoring spontaneous movements.

**Fluorescence microscopy**. After treatments, at least 30 worms per condition, from at least two different experiments were picked and immobilized by 20 mM NaN₃ and washed in M9 buffer onto a 2% agarose pad. Microscopic examinations were carried out by an OLYMPUS CKX53 Fluorescence microscope linked to an OLYMPUS DP74 Cooled color camera. Standard DAPI (Ex/Em 340-390/420IF) and GFP (Ex/Em 469 ± 17.5/525 ± 19.5) filters were used for autofluorescence, an TRITC/Texas Red (Ex/Em 540-550/575-625) filter for Nile Red and a blue-violet IX3-FBVWXL filter (Ex/Em 400-440/460IF) for PA14. Fluorescent signals were acquired with identical signal acquisition settings within experiments. Visualization was performed using the OLYMPUS CellSens v2.3 Imaging software. CTCF (Corrected Total Cell Fluorescence) values of whole animals were calculated by subtracting the background fluorescence from the fluorescence signal of the worm (i.e., the fluorescence intensity of worm/ (mean background fluorescence intensity x area of worm) using ImageJ.

**Pseudomonas aeruginosa PA14 maintenance and infection assays**. PA14 strain was obtained from David W. Wareham (Queen Mary University of London, London, UK). Maintenance and preparation of PA14 were conducted according to[82]. PA14 from overnight cultures was spread on either high osmolarity PGS (pepton-glucose-sorbitol) agar plates for fast killing assays or 140% Bacto-peptone NGM agar plates for mRNA expression assays were prepared according to ref. [38] and incubated at 37 °C for 24 h, then at 23 °C for 6 h. For killing, 30–40 1-day young adults were picked from synchronized populations onto triplicates of PA14-seeded PGS plates. Due to hyperosmolarity, animals were temporarily immobilized for ≈2–3 h. After the restoration of motility, viability was determined by the absence of movement in response to gentle tapping of the plate. Animals that crawled off the agar were censored.

**mRNA expression analysis**. Total RNA isolation was carried out from synchronized adults, using GeneJet RNA Purification Kit (Thermo Scientific). RNA isolates were transcribed into cDNA by RevertAid™ Premium Reverse Transcriptase (Thermo Scientific). Transcripts were diluted into a final volume of 100 μl nuclease-free water and stored at −20 °C. Primers are listed in Supplementary Table 3. act-4 primer sequences are from ref. [83]. qPCR measurements were performed in triplicates with MaximaTM SYBR Green/ROX qPCR Master Mix (Thermo Scientific) in an ABI 7300 Real-time PCR machine using the Comparative Cycle Threshold Method. Relative mRNA levels were normalized to the mRNA level of beta-actin (act-4).

**Statistics and reproducibility**. Statistical analysis was conducted using IBM SPSS Statistics and GraphPad Prism 9.1.2. Kaplan–Meier log-rank tests were carried out to evaluate thermotolerance and PA14 killing assays. Odorant toxicity, paraquat tolerance as well as lawn avoidance were examined by two-way ANOVA with Fisher's LSD post hoc test. qRT-PCR results were analyzed by two-way ANOVA with Fisher's LSD post hoc test prior to normalization to controls. DAF-16::GFP localization was analyzed using a pairwise comparison of a two-tailed Student t-test. Results were extracted from at least three (or two for nuclear DAF-16 nuclear localization in glo-1 mutants) independent experiments and reproducibility was confirmed. Data were expressed as mean ± SEM, where N equals the number of independent experiments. Nalimov test was used to evaluate outlier values within a set of experiments.

**Reporting summary**. Further information on research design is available in the Nature Portfolio Reporting Summary linked to this article.

## Data availability

All data supporting the findings of this study are included in the article and its Supplementary Information. Numerical source data for all charts and graphs can be found in Supplementary Data 1, while those for Figs. 4a, b, and 8a–d can be found in Supplementary Tables 1 and 2 of the Supplementary Information.

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

## Acknowledgements

We thank the *Caenorhabditis* Genetics Center for *C. elegans* strains, Greg Hermann for the *pgp*-2 RNAi bacterial strain, David W. Wareham for the *P. aeruginosa* PA14 strain, and Wormbase for collecting and providing data on *C. elegans*. We are grateful to István Móra and Beatrix Gilányi for technical help, and to other members of the Stress Group for discussions. We thank Norbert Gyöngyösi for his help in statistics. C.S. is thankful for the Merit Prize of the Semmelweis University. This work was funded by grants from the Hungarian Science Foundation (OTKA K116525) to C.S., (OTKA PD142838) to M.S. and (OTKA K131458) to P.C., from the Semmelweis University (STIA_18_M/6800313263, STIA-KFI-2020/132257/AOMBT/2020) and from the Department of Molecular Biology of the Semmelweis University (Baron Munchausen Program 2023/1) to C.S. and by the Thematic Excellence Programme (Tématerületi Kiválósági Program, 2020-4.1.1.-TKP2020, TKP2021-EGA-24) of the Ministry for Innovation and Technology in Hungary, within the framework of the Molecular Biology thematic program of the Semmelweis University to P.C. The funders had no role in study design, data collection, and interpretation, or the decision to submit the work for publication.

## Author contributions

G.H. and C.S. conceived the study. G.H. and C.S. designed the experiments. G.H. and M.S. performed the experiments. G.H., M.S., and C.S. analyzed the data. P.C. and C.S. provided funding. G.H. and C.S. wrote; P.C. and C.S. reviewed the manuscript. All authors read and approved the manuscript.

## Funding

## Competing interests

The authors declare no competing interests.
