## [Peer Review File · Communications Biology]

Reviewers' comments:

Reviewer #1 (Remarks to the Author):

In this paper the authors further expand their idea that the lysosome-related organelles (LROs) promote stress and immune defenses in *C. elegans* and they presented a convincing model for the LRO stress response.

The article provides new insight into the mechanism of how the toxic insults (benzaldehyde, methyl-salicylate and *P. aeruginosa* PA14 infection) are able to induce a deposition of (auto)fluorescent metabolites and an increased burden of *C. elegans* intestinal LROs. Next it was also evidenced, that this in turn stimulates the early expression of LRO-specific membrane transporter and vesicular traffic genes (via yet unidentified factors), and thereby ultimately resulting in LRO adaptation. As the authors outlined, their novel findings illuminate a role for LROs in mounting systemic stress, detoxification and antimicrobial defenses. This process requires a co-operation between ancient and highly conserved processes such as compartmentalization in the endolysosomal system, an ABC membrane transporter and transcriptional stress and innate immune pathways.

In my opinion, this is an excellent paper. This is of sufficient general interest to warrant publication in this journal.

The paper is written clearly and the evidences justifies the conclusions. All experimental protocols are given in sufficient detail.

Reviewer #2 (Remarks to the Author):

Brief Summary

The work by Hajdu et al. addresses the function of gut granules in *C. elegans*. These lysosome-related organelles have been implicated in a number of metabolic processes, including metal and toxin storage. This interesting work shows that gut granules function in protecting *C. elegans* from chemical toxins, heat and oxidative stress, and bacterial pathogens. Moreover, the work identifies transcriptional responses that require gut granules. Together, this work makes a compelling case for a role of gut granules in varied and important physiological processes.

Impressions of the work

Overall, the data is of high quality, is well controlled, and provides convincing support for the major conclusions of the work. The graphical representation and statistical analysis of the work is a strength. The authors show a robust understanding of the literature on *C. elegans* LROs, stress, and pathogens. The authors present a significant amount of data, all of which are central to their questions and conclusions. The work is highly novel, being the first to analyze the regulation of genes involved in LRO formation and function. The work will be of broad interest to the *C. elegans* community as well as those that study the GLO-1 related Rab32/38 and ABC transporters in other organisms, making it a good fit for publication in *Communications Biology*.

Specific comments (numbered)

1. The authors might consider supporting their discussion of the evolutionary origins of LROs by mentioning acidocalcisomes in unicellular organisms, which share functions and biogenesis factors with LROs.
2. On line 43 the authors use malanine when they likely are referring to melanin.
3. On lines 56-57 the wording regarding the function of Rab32/38 and BLOC-3 is unclear.
4. In the experiments presented in Figure 1, were identical fluorescence signal acquisition settings used in control vs BA or N2 vs glo-1(-) conditions? If yes, this should be described in the methods or

figure legend. In addition, it would be helpful to define CTCF units and describe how they were calculated in the methods.

5. In Figure 1B, there is some concern that the addition of Nile Red could increase the green fluorescence, as it can fluoresce in the green. A Control/non-BA condition is not provided to assess this issue. The colocalization of the BA induced fluorescence might be better assessed by localizing gut granules with its red autofluorescence.

6. The data in 1B nicely show colocalization that can only be seen when I really zoom in, for readers it would be helpful to greatly zoom in on one section of the intestine so that individual organelles can be more easily resolved.

7. Please include scale bars in the insets in Fig 1C.

8. In Fig 2E it looks like there was not a significant reduction in lawn avoidance of BA-preconditioned glo-1(-) compared to naïve glo-1(-), yet the authors state on line 183 that there was a partial reduction. This conclusion does not appear to be supported by the data and needs to be changed or addressed more clearly.

9. The authors need to consider and possibly experimentally test the possibility that the behavioral effects that are seen in glo-1(-) mutants could result from LRO independent processes. For example, necrotic cell death of neurons is altered by glo-1(-) <https://doi.org/10.1083/jcb.200511103> and glo-1 expression has not been analyzed for neuronal expression, leaving open the possibility for a role for this Rab in neurons.

10. The suggestion that pre-exposure to BA causes increased Nile Red staining due to increased transport /capacity is only weakly supported by the data (lines 200-203). It could easily be that gut granules accumulate more hydrophobic materials stained by Nile Red in response to BA (due to defects in transport). Assays of fluid phase endocytosis (Rhod-BSA or Rhod-dextran) and organelle diameter would more directly assess changes in transport/capacity.

11. If the investigators have done the analysis, it would be interesting to report the effects of stress, pre-treatment with BA, and loss of gut granules in glo-3(-) on glo-1 mRNA levels, given its key role in LRO biogenesis.

12. The images in Fig 7I do not enable the reader to assess the patterns and localization of fluorescent materials. Larger high resolution images are necessary for the authors to support their claims.

Reviewer #3 (Remarks to the Author):

Hajdu and colleagues present a manuscript where they attempt to assign a function to lysosomal-related organelles in immune and stress response in *C. elegans*. The primary tool employed by these investigators in this study is benzaldehyde, a toxic xenobiotic that increases gut granule autofluorescence. Glo-1 and Glo-3 abrogated the induction of gut granules by BA, as expected. They authors show that pre-exposure to BA provides protection from a subsequent challenge with BA, and show that BA induces the expression of a wide array of stress and immune response genes, albeit subtly (2-8 fold expression for most). In addition, BA can provide some protection from heat stress and paraquat challenge. However, the core conclusion of this paper is that unifying mechanism by which BA promotes stress resistance is through the expansion of lysosomal-related organelles. However, the data supporting this conclusion is only correlative, and multiple alternate explanations exist that could explain their findings, which were not evaluated by the authors. My opinion, which I expand further below, is that the authors have grossly over-interpreted their findings.

1. Fundamentally, the only data that the authors show to support their conclusion that LROs are involved in stress resistance programs is the following: glo mutants suppress autofluorescence and also blunt the BA-induced protection to various stress. However, these two observations could be "true, true, and unrelated." It seems equally plausible that BA, a toxic xenobiotic, activates stress response pathways (e.g., p38, SKN-1, ATFS-1), and gut granule autofluorescence changes to BA treatment as a secondary consequence, rather than a direct effect of LRO biology. The fact that BA-mediated protection is blunted in glo mutants could be secondary to non-specific pleiotropies in these mutant backgrounds, rather than specific changes in the LRO compartment.

2. A direct connection between the LRO and protective stress responses was not established in this paper. Thus, line 24 of the abstract, the core conclusion of the manuscript, "LROs called gut granules promote organismal defenses" is a gross overstatement of the author's findings.

3. Transcription analyses were performed by qRT-PCR on cherry-picked genes, making their biological significance impossible to interpret. Maybe BA treatment increases the transcription of half the genome?

SEMMELWEIS UNIVERSITY

Faculty of Medicine
Department of Molecular Biology

Head of Department:
Prof. MIKLÓS CSALA M.D., Ph.D., D.Sc.

April 18, 2023

Reply to the Reviewers' comments

Please find below our point-by-point reply and the changes made (black font) to the comments of the Reviewers (blue font):

Please note that all line and Figure numbers in our Reply refer to those of the revised article file.

Reviewer #1

In this paper the authors further expand their idea that the lysosome-related organelles (LROs) promote stress and immune defenses in *C. elegans* and they presented a convincing model for the LRO stress response.

The article provides new insight into the mechanism of how the toxic insults (benzaldehyde, methyl-salicylate and *P. aeruginosa* PA14 infection) are able to induce a deposition of (auto)fluorescent metabolites and an increased burden of *C. elegans* intestinal LROs. Next it was also evidenced, that this in turn stimulates the early expression of LRO-specific membrane transporter and vesicular traffic genes (via yet unidentified factors), and thereby ultimately resulting in LRO adaptation. As the authors outlined, their novel findings illuminate a role for LROs in mounting systemic stress, detoxification and antimicrobial defenses. This process requires a co-operation between ancient and highly conserved processes such as compartmentalization in the endolysosomal system, an ABC membrane transporter and transcriptional stress and innate immune pathways.

In my opinion, this is an excellent paper. This is of sufficient general interest to warrant publication in this journal.

The paper is written clearly and the evidences justifies the conclusions. All experimental protocols are given in sufficient detail.

We sincerely thank Reviewer #1 for taking the time to attentively review our manuscript, and for the positive comments and recommendation for publication.

Reviewer #2

Brief Summary

The work by Hajdu et al. addresses the function of gut granules in *C. elegans*. These lysosome-related organelles have been implicated in a number of metabolic processes, including metal and toxin storage. This interesting work shows that gut granules function in protecting *C. elegans* from chemical toxins, heat and oxidative stress, and bacterial pathogens. Moreover, the work identifies

Address: Tűzoltó utca 37-47., H-1094, Budapest, Hungary

Mailing address (P.O.B.): H-1428, Budapest, Pf. 2., Hungary

E-mail: titkarsag.ovi@med.semmelweis-univ.hu; Phone: (+36 1) 266 2615, (+36 1) 459 1500 / ext.: 60100
semmelweis.hu/molekularis-biologia/en

transcriptional responses that require gut granules. Together, this work makes a compelling case for a role of gut granules in varied and important physiological processes.

Impressions of the work

Overall, the data is of high quality, is well controlled, and provides convincing support for the major conclusions of the work. The graphical representation and statistical analysis of the work is a strength. The authors show a robust understanding of the literature on *C. elegans* LROs, stress, and pathogens. The authors present a significant amount of data, all of which are central to their questions and conclusions. The work is highly novel, being the first to analyze the regulation of genes involved in LRO formation and function. The work will be of broad interest to the *C. elegans* community as well as those that study the GLO-1 related Rab32/38 and ABC transporters in other organisms, making it a good fit for publication in *Communications Biology*.

Specific comments (numbered)

1. The authors might consider supporting their discussion of the evolutionary origins of LROs by mentioning acidocalcisomes in unicellular organisms, which share functions and biogenesis factors with LROs.

We sincerely thank Reviewer #2 for the time to evaluate our manuscript, for finding it novel, relevant and sound. We are grateful for the useful comments to improve our study.

We supplemented the Introduction with the inclusion of acidocalcisomes as evolutionary predecessors of LROs (line 56 of the revised MS).

2. On line 43 the authors use malanine when they likely are referring to melanin.

We thankfully corrected the typo (line 48 of the revised MS).

3. On lines 56-57 the wording regarding the function of Rab32/38 and BLOC-3 is unclear.

We are grateful to Reviewer #2 for drawing our attention on the confounding wording. We rephrased the sentence (line 61 of the revised MS).

4. In the experiments presented in Figure 1, were identical fluorescence signal acquisition settings used in control vs BA or N2 vs glo-1(-) conditions? If yes, this should be described in the methods or figure legend. In addition, it would be helpful to define CTCF units and describe how they were calculated in the methods.

We appreciate the comment. Yes, we employed identical acquisition settings for all conditions. Both this and the definition and calculation of CTCF were described in the revised Methods (lines 856-861 of the revised MS).

5. In Figure 1B, there is some concern that the addition of Nile Red could increase the green fluorescence, as it can fluoresce in the green. A Control/non-BA condition is not provided to assess this issue. The colocalization of the BA induced fluorescence might be better assessed by localizing gut granules with its red autofluorescence.

We are indebted to Reviewer #2 for pointing out the lack of a control non-BA condition in Fig. 1B. We included this in the revised Fig. 1B, where we did not detect green fluorescence signal from Nile Red (please compare Figure 1A Control and 1B Nile Red upper left panel). We note that we

Address: Tüzoltó utca 37-47., H-1094, Budapest, Hungary

Mailing address (P.O.B.): H-1428, Budapest, Pf. 2., Hungary

E-mail: titkarsag.ovi@med.semmelweis-univ.hu; **Phone:** (+36 1) 266 2615, (+36 1) 459 1500 / ext.: 60100
semmelweis.hu/molekularis-biologia/en

did not observe BA-induced fluorescence signal in the red channel (please see revised Supplementary Fig. 1A, lines 95-96 of the revised MS), which prevented us to perform colocalization studies comparing the green and red channel signals.

6. The data in 1B nicely show colocalization that can only be seen when I really zoom in, for readers it would be helpful to greatly zoom in on one section of the intestine so that individual organelles can be more easily resolved.

We thank Reviewer #2 for the important comment. We revised Figure 1B and included higher magnification micrographs and insets to visualize individual gut granules.

7. Please include scale bars in the insets in Fig 1C.

We included the scale bars in the insets in Figure 1C.

8. In Fig 2E it looks like there was not a significant reduction in lawn avoidance of BA-preconditioned *glo-1(-)* compared to naïve *glo-1(-)*, yet the authors state on line 183 that there was a partial reduction. This conclusion does not appear to be supported by the data and needs to be changed or addressed more clearly.

We thank for the comment and revised the conclusion of Figure 2E accordingly (line 197 of the revised MS).

9. The authors need to consider and possibly experimentally test the possibility that the behavioral effects that are seen in *glo-1(-)* mutants could result from LRO independent processes. For example, necrotic cell death of neurons is altered by *glo-1(-)* <https://doi.org/10.1083/jcb.200511103> and *glo-1* expression has not been analyzed for neuronal expression, leaving open the possibility for a role for this Rab in neurons.

We thank for this note, cited the suggested reference and included the possibility that the behavioural change might be attributed to a neuronal function of *glo-1* (lines 200-202 of the revised MS).

10. The suggestion that pre-exposure to BA causes increased Nile Red staining due to increased transport /capacity is only weakly supported by the data (lines 200-203). It could easily be that gut granules accumulate more hydrophobic materials stained by Nile Red in response to BA (due to defects in transport). Assays of fluid phase endocytosis (Rhod-BSA or Rhod-dextran) and organelle diameter would more directly assess changes in transport/capacity.

We sincerely thank Reviewer #2 for pointing out weakly supported conclusions regarding a more efficient transport or capacity of LROs. Based on the proposal we monitored the kinetics of Nile Red release from the LROs and found that BA-treated animals exhibited a rapid release of Nile Red from their LROs which became identical to control worms within 30-60 min (revised Fig. 3B). This excludes an unspecific accumulation and transport defect and suggest an increased transport capacity, which is consistent with the transcriptional upregulation of LRO membrane transporters *pgp-2* and *K09C4.5* (revised Figure 3D). We revised the corresponding paragraph accordingly (lines 222-226 of the revised MS).

Although the determination of organelle diameter would provide an important characteristic, we did not observe a robust increase in their size by BA treatment, only a significantly increased red and green fluorescence. The difficulty with these data is that they do not provide direct estimates on organelle size or number, only the quantity of their content. Therefore, we removed the misleading term increased ‘capacity’ which might refer to morphological enlargement and only use increased ‘transport capacity’ or ‘functional capacity’ of LROs from the revised manuscript (see for instance the title of the chapter at lines 211-212 of the revised MS).

11. If the investigators have done the analysis, it would be interesting to report the effects of stress, pre-treatment with BA, and loss of gut granules in *glo-3(-)* on *glo-1* mRNA levels, given its key role in LRO biogenesis.

We thank the important suggestion. We tested the effect of BA exposure and different stress regulator mutations on *glo-1* mRNA levels and found no significant change. These results were included in revised Supplementary Fig. S3E and lines 234-237 of the revised MS. Intriguingly, PA14 infection elevated *glo-1* mRNA level, which was absent in *pgp-2* mutants, suggesting that pathogen stress causes a demand which requires the upregulation of *glo-1* to support LRO biogenesis and a regulatory role of *pgp-2* (see revised Fig. 8H and lines 533 and 536-537 of the revised MS).

12. The images in Fig 7I do not enable the reader to assess the patterns and localization of fluorescent materials. Larger high resolution images are necessary for the authors to support their claims.

We thank Reviewer #2 for the important comment. We included higher magnification micrographs to visualize the fluorescent patterns in revised Figure 8I.

We thank Reviewer #2 again for the constructive comments, and we hope that our reply and the revised manuscript will be recommended for publication in *Communications Biology*.

Reviewer #3

Hajdu and colleagues present a manuscript where they attempt to assign a function to lysosomal-related organelles in immune and stress response in *C. elegans*. The primary tool employed by these investigators in this study is benzaldehyde, a toxic xenobiotic that increases gut granule autofluorescence. *Glo-1* and *Glo-3* abrogated the induction of gut granules by BA, as expected. They authors show that pre-exposure to BA provides protection from a subsequent challenge with BA, and show that BA induces the expression of a wide array of stress and immune response genes, albeit subtly (2-8 fold expression for most). In addition, BA can provide some protection from heat stress and paraquat challenge. However, the core conclusion of this paper is that unifying mechanism by which BA promotes stress resistance is through the expansion of lysosomal-related organelles. However, the data supporting this conclusion is only correlative, and multiple alternate explanations exist that could explain their findings, which were not evaluated by the authors. My opinion, which I expand further below, is that the authors have grossly over-interpreted their findings.

1. Fundamentally, the only data that the authors show to support their conclusion that LROs are

Address: Tüzoltó utca 37-47., H-1094, Budapest, Hungary

Mailing address (P.O.B.): H-1428, Budapest, Pf. 2., Hungary

E-mail: titkarsag.ovi@med.semmelweis-univ.hu; **Phone:** (+36 1) 266 2615, (+36 1) 459 1500 / ext.: 60100

semmelweis.hu/molekularis-biologia/en

involved in stress resistance programs is the following: *glo* mutants suppress autofluorescence and also blunt the BA-induced protection to various stress. However, these two observations could be “true, true, and unrelated.” It seems equally plausible that BA, a toxic xenobiotic, activates stress response pathways (e.g., p38, SKN-1, ATFS-1), and gut granule autofluorescence changes to BA treatment as a secondary consequence, rather than a direct effect of LRO biology. The fact that BA-mediated protection is blunted in *glo* mutants could be secondary to non-specific pleiotropies in these mutant backgrounds, rather than specific changes in the LRO compartment.

We sincerely thank Reviewer #3 for the time to evaluate our manuscript. We are very grateful for pointing out the missing pieces of evidence and the discrepancy between our results and the proposed model.

Based on the Reviewer’s suggestion, we investigated the BA-induced autofluorescence in stress pathway regulator mutants shown or suspected to be involved in the BA-induced stress response. We observed that *daf-16*, *hsf-1*, *pmk-1* and *skn-1*, but not *atfs-1* and *kgl-1*, were required for autofluorescence (revised Fig. 3E and Supplementary Fig. 3D; lines 275-285 and 617-620 of the revised MS). The involvement of other regulator(s) cannot be excluded. These results suggest that the deposition of autofluorescent BA-related material in LROs is a consequence of the activation of several stress pathways, which represent a regulated defensive response against BA toxicity.

We also tested the requirement of these regulators in the BA-induced expression of LRO-associated genes and found that *daf-16* (for *pgp-2*), *hsf-1* (for *pgp-2*) and *pmk-1* (for *glo-3*) were, but *skn-1* was not, required for the expression of LRO mRNAs. Moreover, the expression of *K09C4.5* was affected by none of the above factors, suggesting the involvement of other unknown regulator(s) (revised Fig. 3F-H and Supplementary Fig. 3E,F; lines 286-297 and 640-642 of the revised MS). These findings suggest that the transcriptional upregulation of LRO biogenetic and transport genes upon BA exposure relies on specific, partially identified regulators and appears as an organellar adaptive response to xenobiotic BA stress.

We observed that BA-preconditioning induced protection towards various stresses was compromised in LRO mutants. This raised the question whether LRO mutants might exhibit non-specific pleiotropic effects, i.e. sickness. However, we found that LRO mutants exhibit identical basal physiological tolerance to BA and paraquat and only slightly reduced thermotolerance and pathogen resistance, compared to wild-type (Figs. 4, 6 and 8). Further, although both *pmk-1* and *atfs-1* mutants have lower basal paraquat tolerance, only *pmk-1* failed to respond to BA preconditioning (revised Supplementary Fig. 7E and lines 492-493 of the revised MS). These findings suggest that young adult LRO mutants appear to have quasi-normal physiology, but their response to a preconditioning mild stress is impaired. Moreover, LRO mutants have slightly shorter lifespan and weakened protection against proteotoxicity (Kumar et al Austin Aging Res 2017). We note that stress response mutants (*hsf-1*, *daf-16*, *skn-1*, *pmk-1*) exhibit similar phenotypes, such as shorter lifespan and sensitivity to stresses, including infection. Finally, instead of a compensatory upregulation of other stress responses as a result of sickness, LRO mutations prevented the BA- and PA14-induced activation of specific, but not all transcriptional stress responses (Figs. 5 through 8). Although a mild sickness, i.e. a sensitivity to stress is evident, all the regulated deposition of toxic metabolites, the induction of LRO genes and the expression of other LRO-dependent specific stress responses suggest that LROs regulate organismal stress defenses.

We agree with the Reviewer that the deposition of autofluorescent material in gut granules represent a metabolic process, while the LRO-dependent transcriptional activation of stress and

Address: Tüzoltó utca 37-47., H-1094, Budapest, Hungary

Mailing address (P.O.B.): H-1428, Budapest, Pf. 2., Hungary

E-mail: titkarsag.ovi@med.semmelweis-univ.hu; **Phone:** (+36 1) 266 2615, (+36 1) 459 1500 / ext.: 60100

semmelweis.hu/molekularis-biologia/en

antimicrobial responses might indicate a signaling event, hence, they might be independent processes. Also, the extent of contribution of these processes in the organismal defense and their connection to each other is unclear (see lines 671-673 of the revised MS).

2. A direct connection between the LRO and protective stress responses was not established in this paper. Thus, line 24 of the abstract, the core conclusion of the manuscript, “LROs called gut granules promote organismal defenses” is a gross overstatement of the author’s findings.

We are thankful for the important note. We studied the role of the respective regulators of the BA-induced, LRO-dependent stress and antimicrobial gene expression and found a requirement for *daf-16* (for *hsp-16.2* and *cyp-35B*) and *pmk-1* (for *hsp-70*, *irg-5* and *clcc-60*) and *nhr-86* (for *irg-5*), but not *hsf-1* and *skn-1*. Moreover, the extended nuclear translocation of DAF-16::GFP with a simultaneous inability to activate its target genes in LRO mutants provides evidence that LRO-derived signals are indispensable for the BA-induced expression of stress and immune response genes (Fig. 7 and Supplementary Fig. 7; lines 455-492 and 687-708 of the revised MS). These findings confirm the original conclusion that LROs promote specific organismal defenses and extend it with the demonstration of the extensive cooperation of LROs with cellular stress and immune pathways. We updated the manuscript and the model (revised Fig. 9 and its legend) to reflect a more precise interpretation of the findings.

3. Transcription analyses was performed by qRT-PCR on cherry-picked genes, making their biological significance impossible to interpret. Maybe BA treatment increases the transcription of half the genome?

We highly appreciate the relevant question. It may well be that BA and other toxic agents, such as those of PA14 induce a more comprehensive stress response, which, together with the underlying LRO-dependent mechanisms are subject of a future system-wide research project. In the current study we analyzed the expression of carefully selected marker genes of specific stress responses validated in a plethora of previous studies. Further, besides the induction of specific stress pathways, the biological significance of our findings lies in the induction of adaptive LRO responses (exemplified by *pgp-2* and *glo-3* expression and increased Nile Red transport) as well as the elicitation of organismal defenses to various stresses, including infection (see lines 659-669 of the revised MS).

We are indebted to Reviewer #3 again for the constructive critics and for the thoughtful comments. We hope our reply and the revised manuscript adequately addresses the concerns raised and the study will be recommended for publication in *Communications Biology*.

Sincerely yours,

Csaba Solti, M.D., Ph.D.

Reviewers' comments:

Reviewer #2 (Remarks to the Author):

Review of revised manuscript by Hajdu et al.

My overall impressions of the work have not changed.

I note that the version of the manuscript supplied to me did not have line numbers that matched the authors description in the rebuttal letter making it difficult to assess changes to the manuscript.

Most of my original concerns have been addressed in this improved manuscript.

However, I do still have some minor comments and concerns that relate to major conclusions made by the authors.

1. Prior comment 3. On lines 56-57 the wording regarding the function of Rab32/38 and BLOC-3 is unclear. Response: We rephrased the sentence (line 61 of the revised MS). The description of the relationship between Rab32/38 and BLOC-3 is not correct. BLOC-3 acts as the GEF to activate Rab32/38 and the description would be clearer if there were a description of Rab32/38 function in LRO formation: see <https://www.ncbi.nlm.nih.gov/pmc/articles/PMC4970331/>
2. In Fig 1 the conclusion that induced green autofluorescence localizes to gut granules is not convincing. Without a red channel image in the 1A control/BA it could be that autofluorescence increases in the red channel as well so what is perceived as Nile Red in gut granules is really autofluorescence accumulating somewhere else. The data in Supp Fig 1A suggests that BA does not lead to strong red fluorescence, however the green signal is so weak and the objectives/image acquisition settings are so different in this image it is difficult to be convinced of the authors conclusion that BA does not lead to a significant increase in red autofluorescence signal. While the glo-1(-) result in Fig 1C is consistent with gut granules being the location of the induced autofluorescence it is not enough on its own to make a strong conclusion about the location of autofluorescent material in BA conditions.
3. Scale bars should be included in the insets in Fig 1B and 1C.
4. In Fig 3A,B, E the experiment leading to the data is unclear. Is it the length of time after being exposed to Nile Red or the period of time after feeding ceased that is referred to in the figures?
5. The authors response to my original comment #10 is not convincing. They present experiments in Fig 3A,B, E, which I admit are not very clear to me, that do not directly address anything about transport to and from LROs, they just measure the brightness of red fluorescence, which may not even be from Nile Red (see comment 2). If the signal comes from Nile Red it is unclear how Nile Red gets there or what it stains so conclusions relating to transport or not justified. It is important to keep in mind that Nile Red can move between organelles and accumulates in hydrophobic environments. Without stronger data, the authors should consider removing the conclusion that BA elicits increased functional capacity of *C. elegans* gut granules (line 537).
6. The authors clearly show an increase in autofluorescence upon culturing with BA. However, it is unclear that this is deposition vs chemical changes in pre-existing materials to cause enhanced fluorescence.
7. It is unclear how the data in Supp Fig 1 indicate that the BA response is independent of AA-derived and of glycosylated proteins (104-105). What would the authors expect to see if flu-1 were involved in the creation of the fluorescence and how does the flu-1(-) experiment address glycosylated proteins?

Reviewer #3 (Remarks to the Author):

I appreciate the authors careful attention to the concerns I raised in the initial review of my paper,

and I think the data in this paper present an advance. However, before this paper is formally accepted, I request that the authors use more careful language when describing their results, particularly in the abstract. This is very important in my opinion, because in places, their text does not accurately present their findings, and includes several phrases that are over-interpretations of their findings.

1. Line 29-30. Please delete "consistent with effector-triggered immunity" from the abstract. The authors did not test this, identify bacterial mutant effectors that deplete LROs or show that this is a form of ETI.
2. Line 24. Please be more specific in the sentence "C. elegans LROs promote organismal protection." Such as, "Here we report that RNAi of genes that reduce LRO number leads to enhance susceptibility to multiple stresses, including the toxic xenobiotic benzaldehyde, heat and oxidative stress and pathogen infection"
3. Line 29 Rather than the phrase "involving daf-6 and pmk-1" I ask that the authors state exactly what they showed. Something like "Upregulation of glo-1 dependent genes by benzaldehyde was reduced in daf-16 and pmk-1 dependent mutants". This is important because the authors did not confirm that these pathways were activated (e.g., with western blots for active p38, showing that these mutants are required for their phenotype, genetic epistasis experiments, etc.).
4. Line 31: add "on gene transcription" to the sentence that ends with "benzaldehyde-induced effects."
5. A more accurate conclusion sentence for this abstract is (lines 33-35): "Our study suggests that LROs may play a role in response to systemic stresses and pathogen resistance."
6. Please delete "an organellar stress response" in lines 34, 74, 543. This phrase is non-specific, poorly defined and its meaning is unclear.
7. Lines 77-78 and lines 412-413 and Fig. 7 title: Please state exactly what you show, as stated in the comment under #3. The phrase "cooperates predominantly" and "cooperate" here is not an accurate characterization of the findings.
8. Please be more specific in the title for the results section "LROs promote pathogen defense" Line 456 and Fig. 8 title, such as "RNAi of genes that reduce LRO number leads to enhance susceptibility to P. aeruginosa infection"
9. I would consider deleting the model in Figure 9. It is confusing, difficult to interpret and filled with hypotheses rather than a conclusion of what the author showed.
Other important points to address:
10. Since AWC neurons are known to sense compounds like benzaldehyde, do AWC-ablated animals still have LRO accumulation during BA treatment? Is the accumulation of autofluorescent materials BA-derived or some other host factor not BA-related? The authors should discuss this potential confounder.
11. Although BA treatment induces LRO autofluorescence, it is unclear what constitutes the autofluorescence. Can the authors comment on this?
12. The reference to NHR-86 is confusing (lines 449-451). If BA does indeed activate irg-5 in an NHR-86-dependent manner, then irg-5 should still be significantly upregulated in a pmk-1 mutant. However, this is not the case (Fig. 7E).
13. The authors implicate multiple pathways in the accumulation of autofluorescent material in LROs during BA treatment. In Figure 3E, however, there is one outlier in the N2+BA PC condition that

seems to be much brighter than the rest. Are the other conditions (i.e., pathway mutants) still not significant if this outlier is censored? Authors should also show the original representative images for this figure.

14. In Figure 3F-H, *glo-3* is not significantly suppressed in the *daf-16* mutant and *pqp-2* is not significantly suppressed in the *pmk-1* mutant. Are the genes completely suppressed in a *daf-16/pmk-1* mutant? What is the relationship between these two pathways and the LRO phenotypes?

15. The fast kill PA14 assay is fast, and 80-90% of animals on PGS agar normally die within 4 hrs. However, the authors report that wild-type animals on these fast-kill conditions can live up to 60-70+ hrs. Can the authors comment on this discrepancy?

SEMMELWEIS UNIVERSITY

Faculty of Medicine
Department of Molecular Biology

Head of Department:
Prof. MIKLÓS CSALA M.D., Ph.D., D.Sc.

June 29, 2023

Reply to the Reviewers' comments

Please find below our point-by-point reply and the changes made (black font) to the comments of the Reviewers (blue font):

Please note that all line numbers in our Reply refer to those of the revised tracked article file saved in pdf format.

Reviewer #2

My overall impressions of the work have not changed.

I note that the version of the manuscript supplied to me did not have line numbers that matched the authors description in the rebuttal letter making it difficult to assess changes to the manuscript.

Most of my original concerns have been addressed in this improved manuscript. However, I do still have some minor comments and concerns that relate to major conclusions made by the authors.

1. Prior comment 3. On lines 56-57 the wording regarding the function of Rab32/38 and BLOC-3 is unclear. Response: We rephrased the sentence (line 61 of the revised MS). The description of the relationship between Rab32/38 and BLOC-3 is not correct. BLOC-3 acts as the GEF to activate Rab32/38 and the description would be clearer if there were a description of Rab32/38 function in LRO formation: see <https://www.ncbi.nlm.nih.gov/pmc/articles/PMC4970331/>

We sincerely thank Reviewer #2 for the second round of review and for the time to evaluate our manuscript. We are grateful for the further useful comments to improve our study.

We are sorry for the inconvenience caused by the inaccurate line numbers. We learned that this is inevitable in doc format. Therefore, we also submitted the tracked manuscript in pdf and will refer to the line numbers of that file.

We apologize for the erroneous description of BLOC-3 function and thank for the suggestion. We corrected it, described the function of RAb32 and BLOC-3 in melanosome formation and also cited the reference suggested (lines 64-67 of the revised MS pdf).

2. In Fig 1 the conclusion that induced green autofluorescence localizes to gut granules is not convincing. Without a red channel image in the 1A control/BA it could be that autofluorescence increases in the red channel as well so what is perceived as Nile Red in gut granules is really autofluorescence accumulating somewhere else. The data in Supp Fig 1A suggests that BA does not lead to strong red fluorescence, however the green signal is so weak and the objectives/image

Address: Tüzoltó utca 37-47., H-1094, Budapest, Hungary

Mailing address (P.O.B.): H-1428, Budapest, Pf. 2., Hungary

E-mail: titkarsag.ovi@med.semmelweis-univ.hu; Phone: (+36 1) 266 2615, (+36 1) 459 1500 / ext.: 60100
semmelweis.hu/molekularis-biologia/en

acquisition settings are so different in this image it is difficult to be convinced of the authors conclusion that BA does not lead to a significant increase in red autofluorescence signal. While the *glo-1(-)* result in Fig 1C is consistent with gut granules being the location of the induced autofluorescence it is not enough on its own to make a strong conclusion about the location of autofluorescent material in BA conditions.

We thank the Reviewer for the important argument, which we agree with. Although we never observed significant red signal in none of the previous experiments, we performed new experiments and made fluorescent micrographs at both the green and red channels, using the same image acquisition settings (new Fig. S1A). This result demonstrates that BA exposure does not elicit considerable red fluorescence. To provide further evidence that the BA-induced material indeed accumulates in gut granules we simultaneously treated N2 and *glo-1* mutant worms with BA and Nile Red (new Fig S2B). The lack of green and red signal in *glo-1* shows that both BA and Nile Red appears only in LROs (see also reply to comment 5).

3. Scale bars should be included in the insets in Fig 1B and 1C.

We thankfully received the comment and fixed the error in Figs 1B and 1C.

4. In Fig 3A,B, E the experiment leading to the data is unclear. Is it the length of time after being exposed to Nile Red or the period of time after feeding ceased that is referred to in the figures?

We thank for drawing our attention to the unclear legends. In panel A, it is Nile Red accumulation (i.e. the various duration after being exposed to Nile Red), in B it is Nile Red release (at the indicated times after Nile Red feeding ceased), whereas in E it is the BA-induced green autofluorescence, where we did not employ Nile Red. We clarified the legend accordingly.

5. The authors response to my original comment #10 is not convincing. They present experiments in Fig 3A,B, E, which I admit are not very clear to me, that do not directly address anything about transport to and from LROs, they just measure the brightness of red fluorescence, which may not even be from Nile Red (see comment 2). If the signal comes from Nile Red it is unclear how Nile Red gets there or what it stains so conclusions relating to transport or not justified. It is important to keep in mind that Nile Red can move between organelles and accumulates in hydrophobic environments. Without stronger data, the authors should consider removing the conclusion that BA elicits increased functional capacity of *C. elegans* gut granules (line 537).

We are thankful for the useful comment. We understand that in light of the concerns raised about the potential red fluorescence of BA (comment 2) and the unclear legend of Fig 3A, B, E (comment 4) it was arguable that we measured Nile Red uptake into LROs. Although we showed in the previous manuscript that *pgp-2* RNAi almost entirely eliminated both BA autofluorescence and Nile Red fluorescence (Fig. S6A and B), the remaining signal might be due to incomplete RNAi penetrance and/or persisting embryonic LROs. Therefore, to specifically address this question, we performed Nile Red feeding experiments in control and BA-treated wildtype N2 and *glo-1* mutants. Whereas we observed the LRO-like granular red fluorescence in N2, we did not detect red fluorescence in *glo-1* worms lacking LROs (new Fig. S1B). We note that our results are consistent with and confirm those of Gary Ruvkun's group that LROs are the site that specifically accumulate Nile Red in a regulated manner (among others, ref 10, Soukas et al, PLoS Genet 2013). Based on their and our results, we conclude that Figs. 3A and S3A show the kinetics of Nile Red

Address: Tüzoltó utca 37-47., H-1094, Budapest, Hungary

Mailing address (P.O.B.): H-1428, Budapest, Pf. 2., Hungary

E-mail: titkarsag.ovi@med.semmelweis-univ.hu; **Phone:** (+36 1) 266 2615, (+36 1) 459 1500 / ext.: 60100
semmelweis.hu/molekularis-biologia/en

accumulation in LROs during Nile Red feeding and Fig 3B shows Nile Red release from LROs after Nile Red feeding ceased. They together indicate that LROs increase their specific Nile Red transport, suggesting an increased functional capacity of LROs in response to BA exposure.

6. The authors clearly show an increase in autofluorescence upon culturing with BA. However, it is unclear that this is deposition vs chemical changes in pre-existing materials to cause enhanced fluorescence.

We agree with the Reviewer that currently it is unclear whether it is a deposition of autofluorescent (BA-related) material or a chemical modification of pre-existing material in response to BA or MS. We modified the corresponding texts to reflect this (lines 144-145, 579 and 597-601 of the revised MS pdf).

7. It is unclear how the data in Supp Fig 1 indicate that the BA response is independent of AA-derived and of glycosylated proteins (104-105). What would the authors expect to see if *flu-1* were involved in the creation of the fluorescence and how does the *flu-1(-)* experiment address glycosylated proteins?

We thank the Reviewer for the comment. Indeed, we have no evidence on the link between BA-autofluorescence and glycosylated proteins, therefore we removed these sentences. The observation that BA-induced autofluorescence is maintained in the AA-deficient *kynu-1* mutant (Fig. S1C) suggests that BA does not stimulate AA formation and the fluorescent material is independent of AA metabolism. We modified the sentence accordingly (lines 123-124 of the revised MS pdf). We appreciate pointing out a different interpretation of the role of *flu-1* in BA-induced autofluorescence and protection against BA toxicity. As our work is focusing on the role of LROs, we included this interpretation in the Results (lines 193-195 of the revised MS pdf) and modified the title of Fig S2.

We thank Reviewer #2 again for the constructive comments, and we hope that our reply answers the questions and the revised manuscript will be recommended for publication in *Communications Biology*.

Reviewer #3

I appreciate the authors' careful attention to the concerns I raised in the initial review of my paper, and I think the data in this paper present an advance. However, before this paper is formally accepted, I request that the authors use more careful language when describing their results, particularly in the abstract. This is very important in my opinion, because in places, their text does not accurately present their findings, and includes several phrases that are over-interpretations of their findings.

1. Line 29-30. Please delete "consistent with effector-triggered immunity" from the abstract. The authors did not test this, identify bacterial mutant effectors that deplete LROs or show that this is a form of ETI.

We sincerely thank Reviewer #3 the second round of review and for the time to evaluate our manuscript. We appreciate the feedback that the revised study represents a progress and for the requirement of more accuracy.

Address: Tüzoltó utca 37-47., H-1094, Budapest, Hungary

Mailing address (P.O.B.): H-1428, Budapest, Pf. 2., Hungary

E-mail: titkarsag.ovi@med.semmelweis-univ.hu; **Phone:** (+36 1) 266 2615, (+36 1) 459 1500 / ext.: 60100
semmelweis.hu/molekularis-biologia/en

We thankfully removed ETI from the abstract.

2. Line 24. Please be more specific in the sentence “*C. elegans* LROs promote organismal protection.” Such as, “Here we report that RNAi of genes that reduce LRO number leads to enhance susceptibility to multiple stresses, including the toxic xenobiotic benzaldehyde, heat and oxidative stress and pathogen infection”

We agree with the Reviewer that organismal protection is unspecific. However, we detail the specific findings after this sentence. Hence, we corrected the abstract: “Here, we report that *C. elegans* LROs (gut granules) promote organismal defenses against various stresses.”

3. Line 29 Rather than the phrase “involving *daf-6* and *pmk-1*” I ask that the authors state exactly what they showed. Something like “Upregulation of *glo-1* dependent genes by benzaldehyde was reduced in *daf-16* and *pmk-1* dependent mutants”. This is important because the authors did not confirm that these pathways were activated (e.g., with western blots for active p38, showing that these mutants are required for their phenotype, genetic epistasis experiments, etc.).

We thank for pointing out the inaccuracy. We corrected the sentence to show only the requirement of these genes: „which requires *daf-16*/FOXO and/or *pmk-1*/p38MAPK”.

4. Line 31: add “on gene transcription” to the sentence that ends with “benzaldehyde-induced effects.”

We corrected the sentence, including the *pgp-2*-dependent effects on stress resistance and due to the 150-word limit of the abstract, we combined it with the previous one on *glo-1*: “We find that toxic benzaldehyde exposure induces LRO autofluorescence, stimulates the expression of LRO-specific genes, enhances LRO transport capacity and tolerance to benzaldehyde, heat and oxidative stresses, which are impaired in *glo-1*/Rab32 and *pgp-2* ABC transporter LRO biogenesis mutants.”

5. A more accurate conclusion sentence for this abstract is (lines 33-35): “Our study suggests that LROs may play a role in response to systemic stresses and pathogen resistance.”

We replaced the concluding sentence: „Our study suggests that LROs may play a role in systemic responses to stresses and in pathogen resistance.”

6. Please delete “an organellar stress response” in lines 34, 74, 543. This phrase is non-specific, poorly defined and its meaning is unclear.

We agree and deleted the term.

7. Lines 77-78 and lines 412-413 and Fig. 7 title: Please state exactly what you show, as stated in the comment under #3. The phrase “cooperates predominantly” and “cooperate” here is not an accurate characterization of the findings.

We thankfully corrected the sentences: „The LRO-associated expression of stress genes mainly requires the *daf-16* FOXO and/or the *pmk-1* p38 orthologs and yet unknown factors.” (lines 93-94) “LRO-dependent stress and immune responses require *daf-16* and *pmk-1*” (line 449-450 of the revised MS pdf and Fig. 7 title)

8. Please be more specific in the title for the results section “LROs promote pathogen defense” Line 456 and Fig. 8 title, such as “RNAi of genes that reduce LRO number leads to enhance susceptibility to *P. aeruginosa* infection”

We thank for the suggestion and corrected the sentences: „LROs are required for resistance against *P. aeruginosa* infection”

9. I would consider deleting the model in Figure 9. It is confusing, difficult to interpret and filled with hypotheses rather than a conclusion of what the author showed.

We agree and removed Figure 9.

Other important points to address:

10. Since AWC neurons are known to sense compounds like benzaldehyde, do AWC-ablated animals still have LRO accumulation during BA treatment? Is the accumulation of autofluorescent materials BA-derived or some other host factor not BA-related? The authors should discuss this potential confounder.

We thank for these comments. We agree that BA at low concentration is sensed by the AWC neuron and elicits attractive behavior, where it does not induce LRO autofluorescence. However, BA at high concentration is toxic and repulsive (Hajdú et al BMC Biol 2021). This also happens in the AWC chemosensory *odr-3* mutants (Nuttley et al Learn Mem 2001, Hajdú and Sóti, unpublished). Hence, although the contribution of neural pathways might not be excluded, it might require further studies beyond the scope of this work. Indeed, it is yet unclear whether the observed autofluorescent signal is due to a deposition of BA-related material or a chemical modification of pre-existing material in response to BA or MS. We modified the corresponding texts to reflect this (lines 144-145, 579 and 597-601 of the revised MS pdf).

11. Although BA treatment induces LRO autofluorescence, it is unclear what constitutes the autofluorescence. Can the authors comment on this?

As explained in the previous point, the nature of the autofluorescent material is unknown and remains to be identified. The finding that benzaldehyde and methyl salicylate, compounds that possess similar aromatic chemical structure, both induce LRO autofluorescence argues for chemical structure-specific phenomenon, which might be related to the detoxification. Another hypothesis is that the LROs are sites for ascaroside synthesis, which also utilizes anthranilate and benzoic acid derivatives and might exhibit autofluorescent properties at excessive doses (see Ref. 25 Le et al eLife 2020 and lines 597-601 and 715-720 of the revised MS pdf).

12. The reference to NHR-86 is confusing (lines 449-451). If BA does indeed activate *irg-5* in an NHR-86-dependent manner, then *irg-5* should still be significantly upregulated in a *pmk-1* mutant. However, this is not the case (Fig. 7E).

We thank for pointing out the confusion regarding the roles of NHR-86 and PMK-1 in *irg-5* regulation. Please note that the basal expression of *irg-5* such as other NHR-86 immune effector targets, requires the canonical p38 MAPK PMK-1 immune pathway (see Ref 49, Peterson et al PLoS Genet 2019). Our findings show a similar, *pmk-1*- and *nhr-86*-dependent induction of *irg-5* in response to BA confirming that both regulators are indispensable for the activation (Fig. 7E and

Supplementary Fig. 7D). We hope this resolves the concern. To clarify this for the readers, we rephrased the sentences (lines 489-492 and 712-715 of the revised MS pdf).

13. The authors implicate multiple pathways in the accumulation of autofluorescent material in LROs during BA treatment. In Figure 3E, however, there is one outlier in the N2+BA PC condition that seems to be much brighter than the rest. Are the other conditions (i.e., pathway mutants) still not significant if this outlier is censored? Authors should also show the original representative images for this figure.

We thank for this point. Although that point seems to be an outlier, it is not an outlier according to the Nalimov test. Therefore, we did not remove this point. We note that even if this is censored, the difference between the N2 control and BA remains significant, whereas that between the mutants' control and BA remains insignificant. As requested, we present the representative images in new Fig. S3G.

14. In Figure 3F-H, *glo-3* is not significantly suppressed in the *daf-16* mutant and *pqp-2* is not significantly suppressed in the *pmk-1* mutant. Are the genes completely suppressed in a *daf-16/pmk-1* mutant? What is the relationship between these two pathways and the LRO phenotypes? We appreciate the relevant question. It is possible that these genes may be completely suppressed in a double mutant. However, the expression of *K09C4.5* is not affected by *daf-16* or *pmk-1*. Other LRO phenotypes, such as autofluorescence, are affected by *hsf-1*. Hence, we speculate that the LRO genes are controlled by known and yet unknown factors. The elucidation of this regulation and the relationship between *daf-16*, *pmk-1* and other regulators in determining LRO phenotypes is an intriguing subject of a subsequent study. We reformulated the Discussion to include these ideas (line 645-648 of the revised MS pdf).

15. The fast kill PA14 assay is fast, and 80-90% of animals on PGS agar normally die within 4 hrs. However, the authors report that wild-type animals on these fast-kill conditions can live up to 60-70+ hrs. Can the authors comment on this discrepancy?

We thankful for pointing out the apparent discrepancy. Fast killing of L4 larvae by PA14 is indeed happens within 4 hours. However, fast killing of 1-day young adults used in our study requires longer time similar to what is reported from the Ausubel lab (see Tan et al PNAS 1999). We also added an explanatory sentence to the manuscript (lines 507-508 of the revised MS pdf).

We thank Reviewer #3 again for the constructive comments. We hope our reply answers the questions and the revised manuscript will be recommended for publication in *Communications Biology*.

Sincerely yours,

Csaba Söti, M.D., Ph.D.

REVIEWERS' COMMENTS:

Reviewer #2 (Remarks to the Author):

The revised manuscript addresses my prior concerns. As it stands, the revised manuscript presents important new insights into the function of LROs in *C. elegans*.

Reviewer #3 (Remarks to the Author):

This study is a resubmission from Hajdu and colleagues which ascribes a role for lysosome-related organelles in the model organism *C. elegans* in response to multiple stresses. The manuscript is improved, however a few changes are still necessary:

Major comments:

1. please remove "effector-triggered immunity" from the keywords
2. Lines 131-132: The authors hypothesized that "the sequestration of BA-derived autofluorescent material by LROs might contribute to the organismal defense against BA toxicity." Might it also be possible, given that LROs may express enzymes, that it is not the sequestration but the direct breakdown or conversion of BA to other metabolites that offers protection? Does the BA-induced autofluorescence decrease over time? Please comment.
3. Lines 445 – 446: It is unclear how the authors reached the hypothesis that LROs would "interfere with the activation of DAF-16-dependent genes." Do the authors suggest that the increased DAF-16 signal in the nucleus (Fig. 7C,D) is compensatory and not indicative of increased function (i.e., increased transcription)? In Fig. 6F, the authors showed that loss of *pgp-2* suppressed BA-induced *cyp-35B*, which is *daf-16*-dependent (Fig. 7A). Please elaborate and clarify since an increase in nuclear expression of DAF-16 is often associated with increased transcription of *daf-16*-dependent genes.

Minor revisions:

1. Lines 38-40: please provide references for these statements
2. Throughout the manuscript, please replace "undiluted" compound (e.g., BA/DA/MS) with the concentration or specify the undiluted concentration in the "Methods" or figure legends.
3. Lines 449-451: include figure reference

SEMMELWEIS UNIVERSITY

Faculty of Medicine
Department of Molecular Biology

Head of Department:
Prof. MIKLÓS CSALA M.D., Ph.D., D.Sc.

August 4, 2023

Reply to the Reviewers' comments

Please find below our point-by-point reply and the changes made (black font) to the comments of the Reviewers (blue font):

Please note that all line numbers in our Reply refer to those of the revised tracked article file.

Reviewer #2

The revised manuscript addresses my prior concerns. As it stands, the revised manuscript presents important new insights into the function of LROs in *C. elegans*.

We are indebted to Reviewer #2 for the third round of review and for the time to evaluate our manuscript. We are grateful for the recommendation for publication in *Communications Biology*.

Reviewer #3

1. please remove “effector-triggered immunity” from the keywords

We deleted the term from the keywords.

2. Lines 131-132: The authors hypothesized that “the sequestration of BA-derived autofluorescent material by LROs might contribute to the organismal defense against BA toxicity.” Might it also be possible, given that LROs may express enzymes, that it is not the sequestration but the direct breakdown or conversion of BA to other metabolites that offers protection? Does the BA-induced autofluorescence decrease over time? Please comment.

We agree with the idea and based on the previous review we have already expressed it in the Discussion (lines 752-758). Now, we also included these possibilities in the abovementioned sentence: “the sequestration, breakdown or conversion of BA to autofluorescent material by LROs might contribute to the organismal defense against BA toxicity.” (lines 152-154)

According to our unpublished data, the BA-induced autofluorescence does not decrease over a 24-hr period, the time-span of our experiments. This is an exciting question, but we did not follow it at longer timescales.

3. Lines 445 – 446: It is unclear how the authors reached the hypothesis that LROs would “interfere with the activation of DAF-16-dependent genes.” Do the authors suggest that the increased DAF-16 signal in the nucleus (Fig. 7C,D) is compensatory and not indicative of increased function (i.e., increased transcription)? In Fig. 6F, the authors showed that loss of *pgp-2* suppressed BA-induced *cyp-35B*, which is *daf-16*-dependent (Fig. 7A). Please elaborate and clarify since an increase in

Address: Tüzoltó utca 37-47., H-1094, Budapest, Hungary

Mailing address (P.O.B.): H-1428, Budapest, Pf. 2., Hungary

E-mail: titkarsag.ovi@med.semmelweis-univ.hu; Phone: (+36 1) 266 2615, (+36 1) 459 1500 / ext.: 60100
semmelweis.hu/molekularis-biologia/en

nuclear expression of DAF-16 is often associated with increased transcription of daf-16-dependent genes.

We thank the Reviewer for pointing out the ambiguity of our presentation of DAF-16 dependent transcription. We exactly meant to propose a compensatory nuclear retention of DAF-16 in the absence of efficient transactivation in LRO mutants. We clarified the corresponding sentences in the manuscript (lines 591-597).

Minor revisions:

1. Lines 38-40: please provide references for these statements

We appreciate the comment and provided the references to these statements (line 46).

2. Throughout the manuscript, please replace “undiluted” compound (e.g., BA/DA/MS) with the concentration or specify the undiluted concentration in the “Methods” or figure legends.

We extended the “Methods” with the advised description: undiluted, 100% (v/v) (line 923).

3. Lines 449-451: include figure reference

We thankfully added the figure reference (Figs. 5a, e, 6e-f) (line 592).

We thank Reviewer #3 again to take the persistent effort for constructive comments. We sincerely hope our reply will be accepted and the revised manuscript will be recommended for publication in *Communications Biology*.

Sincerely yours,

Csaba Solti, M.D., Ph.D.